

# New water fractions and their relationships to climate and catchment properties across Alpine rivers

Marius G. Floriancic[1,2], Michael P. Stockinger[3], James W. Kirchner[1,4], Christine Stumpp[3]

[1] Dept. of Environmental Systems Science, ETH Zürich, Zürich, Switzerland

[2] Dept. of Civil, Environmental and Geomatic Engineering ETH Zürich, Zürich, Switzerland

[3] Department of Water, Atmosphere and Environment, Institute of Soil Physics and Rural Water Management,

University of Natural Resources and Life Sciences, Muthgasse 18, 1190 Vienna, Austria

[4] Swiss Federal Research Institute WSL, Birmensdorf, Switzerland

*Correspondence to*: Marius G. Floriancic (floriancic@ifu.baug.ethz.ch)

**Abstract.** The Alps are a key water resource for central Europe, providing water for drinking, agriculture, and hydropower production. Thus, understanding runoff generation processes of Alpine streams is important for sustainable water management. It is currently unclear how much streamflow is derived from old water stored in the subsurface, versus more recent precipitation that reaches the stream via near-surface quick flow processes. It is also unclear how this partitioning varies across different Alpine catchments in response to hydroclimatic forcing and catchment characteristics. Here, we use stable water isotope time series in precipitation and streamflow to quantify the young water fractions $F_{yw}$ (i.e., the fraction of water younger than approximately 2-3 months) and new water fractions $F_{new}$ (here, the fraction of water younger than one month) in streamflow from 32 Alpine catchments. We contrast these measures of water age between summer and winter and between wet and dry periods, and correlate them with hydroclimatic variables and physical catchment properties.

New water fractions varied from 9.6 % in rainfall-dominated catchments to 3.5 % in snow-dominated catchments (mean across all catchments = 7.1 %). Young water fractions were approximately twice as large (reflecting their longer time scale), varying from 17.6 % in rainfall-dominated catchments to 10.1 % in snow-dominated catchments (mean across all catchments = 14.3 %). New water fractions were negatively correlated with catchment size (Spearman rank correlation $r_S = -0.38$), q95 baseflow ($r_S = -0.36$), catchment elevation ($r_S = -0.37$), total catchment relief ($r_S = -0.59$), and the fraction of slopes steeper 40° ($r_S = -0.48$). Large new water fractions, implying faster transmission of precipitation to streamflow, are more prevalent in small catchments, at low elevations, with small elevation gradients, and with large forest cover ($r_S = 0.36$). New water fractions averaged 3.3 % following dry antecedent conditions, compared to 9.3 % after wet antecedent conditions. Our results quantify how hydroclimatic and physical drivers shape the partitioning of old and new waters across the Alps, thus indicating which landscapes transmit recent precipitation more readily to streamflow, and which landscapes tend to retain





water over longer periods. Our results further illustrate how new water fractions may find relationships that remained invisible with young water fractions.

# 1.       1 Introduction

The Alps are often referred to as the "water tower of Europe", as they contribute disproportionally high fractions of streamflow of European rivers (Weingartner et al., 2007). They provide water for agriculture, domestic use, and hydropower production, not only in the Alpine region, but also for approximately 170 million people living in the downstream basins (Mastrotheodoros *et al.*, 2020). Thus, it is important to understand the origins of streamflow in Alpine rivers and how they might change in future climates (Briffa et al., 2009). So far, little is known about the transport and storage of the waters that

become Alpine river flow, i.e., to what extent streamflow originates from old water stored in the subsurface, and to what extent streamflow consist of more recent precipitation reaching the stream via near-surface quick flow processes. Across the Alps, contributions from both slow subsurface flow and fast surface or near-surface flow processes are poorly understood (Hayashi, 2019). It is likewise unclear how these slow and fast flow processes, and thus the partitioning of old and new water in Alpine streams, are related to hydroclimatic forcing and physical catchment characteristics across different Alpine

catchments.

Stable water isotopes are essential tools for estimating the contribution of different sources to streamflow and for assessing how this source partitioning varies with precipitation characteristics and catchment wetness conditions (Segura et al., 2012). Stable water isotopes have been measured in many catchments worldwide, and data compilations are available for the globe,

e.g., the Global Network of Isotopes in Rivers GNIR (IAEA, 2019a) and the Global Network of Isotopes in Precipitation GNIP (IAEA, 2019b), and for some regions and countries, e.g. for Switzerland (Staudinger *et al.*, 2020), as well as for individual intensively studied catchments (e.g., Hubbard Brook, Plynlimon, Alptal). Although most multi-catchment time series have been sampled at low temporal frequency, they can nonetheless be used to assess the mixture of streamflow sources on time scales similar to their sampling intervals. They can thus yield insights into how long it takes until

precipitation becomes streamflow or, in turn, how much streamflow is coming from recent precipitation versus from water that has been stored in the catchment over longer time scales (Hrachowitz et al., 2009).

Many previous studies assessed the fraction of "event" water by hydrograph separation using two-component mixing models (Klaus and McDonnell, 2013). Prior to the widespread use of stable water isotopes and other conservative tracers, it was

assumed that stormflow mostly originated from recent rainfall that travelled rapidly to the stream via overland flow or preferential subsurface flowpaths (Kirchner *et al.*, 2023). This conceptual model was overthrown by tracer studies that showed that although stream discharge responds quickly to rainfall inputs, recent precipitation makes up only a small fraction of stormflow (Sklash and Farvolden, 1979; Neal and Rosier, 1990; McDonnell and Beven, 2014), a phenomenon

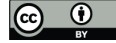


often referred to as the "old water paradox" (Kirchner, 2003; McDonnell *et al.*, 2010). Instead of flowing rapidly to the
stream, most precipitation instead infiltrates and mobilizes older water from subsurface storage (von Freyberg *et al.*, 2017;
von Freyberg *et al.*, 2018). Thus, subsurface storage supplies a large fraction of streamflow, not only during baseflow
conditions but also during high flow events (Browne, 1981; Fleckenstein *et al.*, 2006; Floriancic *et al.*, 2022).

Although aquifer waters can be very old (Gleeson *et al.*, 2016; Jasechko *et al.*, 2017), streamwaters derived from them are
typically much younger (Berghuijs and Kirchner, 2017). The explanation for this apparent paradox is that hydraulic
conductivities in aquifers vary by orders of magnitude (Gleeson et al., 2011b), with the faster flowpaths transmitting
disproportionally more water, which, because it flows faster, is younger than the waters left behind in the slower flowpaths
(Berghuijs and Kirchner, 2017; Kirchner *et al.*, 2023). In global-scale syntheses, Jasechko *et al.* (2016, 2017) found although
most groundwaters are dominated by fossil waters, 25% of global streamflow is younger than 1.5 - 3 months. According to
Jasechko *et al.* (2016), rivers draining mountainous regions tend to have smaller fractions of young water than rivers
draining flatter terrain. Jasechko *et al.* (2016) explain this finding by arguing that steeper areas allow for deeper vertical
infiltration and thus a greater predominance of deeper, slower flowpaths. However, it remains unclear whether the
association between steep terrain and young water fractions is consistent across the European Alps and to what extent these
results are influenced by other hydroclimatic variables and physical catchment properties.


Recent studies assessed the impact of catchment properties on the fraction of young water, defined as the fraction of
streamflow that is younger than approximately 2-3 months, which can be inferred from the amplitude of seasonal tracer
cycles in precipitation and streamflow (Kirchner 2016a; 2016b). For example, von Freyberg *et al.* (2018) found that young
water fractions tend to be smaller in steeper terrain, and Ceperley *et al.* (2020) found a decrease in young water fractions
above an elevation of 1500 m asl. across Swiss & Italian Alpine catchments. Von Freyberg et al. (2018) also found that
higher fractions of young water were associated with higher antecedent catchment wetness, as well as with hydro-climatic
factors and catchment characteristics that favor faster transmission of waters to the stream (wet climates, low subsurface
permeability, high drainage density). Stockinger *et al.* (2019) found that young water fractions were not related to climate
but inversely related to the ratio of average discharge to average precipitation. Catchment area has previously been identified
as a major control on catchment mean transit time (DeWalle *et al.*, 1995; Soulsby *et al.*, 2000), but has not been found to be
significantly related to young water fractions (von Freyberg *et al.,* 2018).

Gentile *et al.* (2023) have argued that young water fractions should depend on the fraction of catchment area covered by
unconsolidated debris deposits and the fraction of baseflow, and von Freyberg *et al.* (2018) found a significant positive
correlation between young water fractions and the fraction of the catchment covered by forests. Previous studies of young
water fractions in the Alpine region have primarily focused on small headwater catchments (Gentile *et al.*, 2023; von
Freyberg *et al.*, 2018; Ceperley *et al.,* 2020) and did not include larger downstream basins. Whereas young water fractions





have been widely used to assess how hydroclimatic and physiographic properties shape catchment transport, new water fractions have thus far remained under-exploited for this purpose. To date, no studies have systematically linked new water
fractions to hydroclimatic drivers and physical catchment properties, using datasets that include both headwater catchments and larger downstream basins.

For this study we compiled time series of streamflow, precipitation, stable water isotopes for 32 catchments across the Austrian and Swiss Alps, spanning a wide range of catchment sizes and elevation gradients. We analyzed these time series
using two recently developed methods for assessing the relative proportions of younger and older water in streamflow, young water fractions $F_{yw}$ (Kirchner 2016a; 2016b; von Freyberg *et al.*, 2017) and new water fractions $F_{new}$ (Kirchner 2019; Knapp *et al.*, 2019; Kirchner and Knapp 2020), to address the following research questions:

-        How much new and young water can be found in Alpine rivers?


-        How do new water fractions vary between different wetness conditions and seasons?

-        How do new water fractions vary with precipitation intensity across Alpine rivers?

-        How do new water fractions propagate downstream from headwater catchments to the large basins of the Danube and Rhine catchments?

-        Which hydroclimatic variables (climate and streamflow response) and physical catchment properties (topography, lithology, landuse) are associated with larger or smaller fractions of new water?

**2.        Methods and Available Data**

**2.1. Precipitation and Streamflow Data**

The analysis is based on 32 Austrian and Swiss Alpine catchments for which streamflow isotope data were available (see Figure 1 and Table 1). Daily discharge time series for 12 of the 20 Swiss sites were obtained from the CH-IRP database
(Staudinger *et al.*, 2020). Discharge time series for the remaining 8 Swiss sites and for all 12 Austrian sites were obtained from the Federal Office of the Environment "Hydrological Data and forecasts" database (FOEN, 2022a) and the "Hydrographisches Jahrbuch" contained in the WISA database (Umweltbundesamt, 2022a), respectively. Daily catchment averages of precipitation for all 32 sites (12 WISA, 8 FOEN, 12 CH-IRP) were obtained from the gridded precipitation dataset E-OBS (version 20.0e) at 0.1-degree resolution covering the period 1980–2014 (Cornes *et al.*, 2018). For this



purpose, the catchment boundaries for all 32 gauging stations were extracted from the Copernicus EU-DEM v1.1 at 25 m resolution using the ArcMap 10.6 Spatial Analyst toolbox.

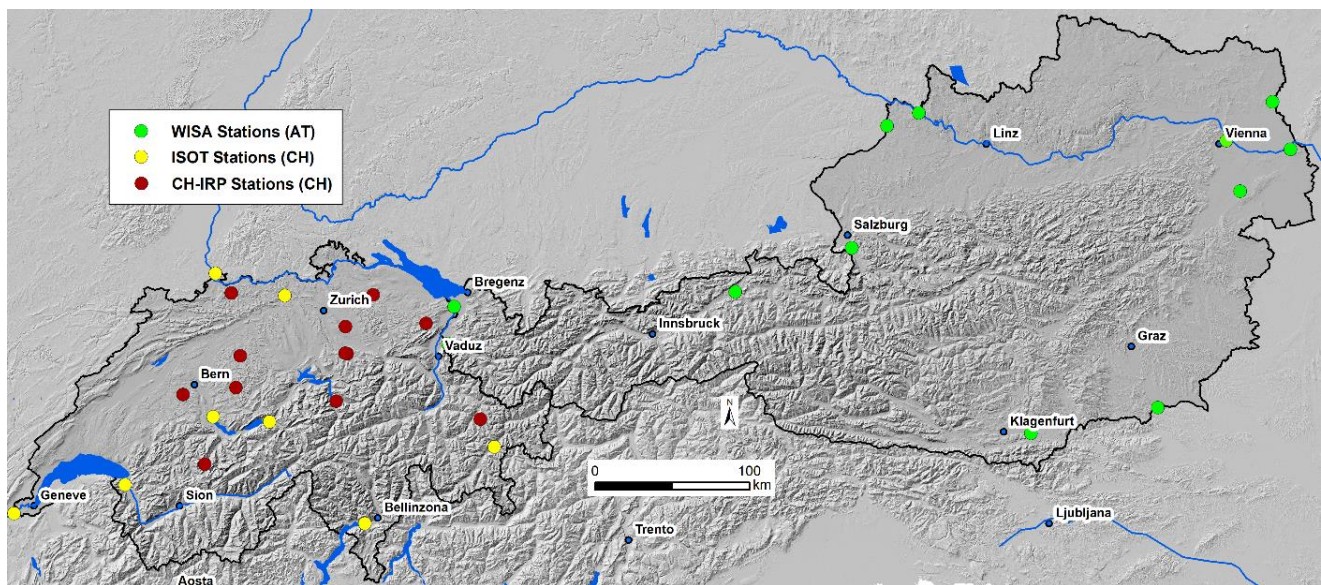

**Figure 1: Location of the streamflow isotope sampling sites in Austria (WISA database – green markers) and**
**Switzerland (ISOT database – yellow markers, CH-IRP database – red markers). The hillshade in this map is based on the EU-DEM v1.1. available through funding by the European Union.**

### 2.2. Isotope Data

We compiled streamflow and precipitation isotope data for 32 catchments across the Austrian and Swiss Alps (see Figure 1 and Table 1). Monthly streamflow isotopes for the 12 Austrian sites were obtained from the WISA "H2O Fachdatenbank" database (Umweltbundesamt, 2022b) and for 8 Swiss stations from the NAQUA ISOT ("Nationalen Grundwasserbeobachtung – Isotopendaten") database (FOEN, 2022b). Streamflow isotopes for 12 additional stations across the Swiss Alps were obtained from the CH-IRP database (Staudinger *et al.*, 2020). Many of the study catchments lack direct

measurements of precipitation isotopes within the catchment boundaries. In all but the largest catchments where direct measurements are available, they are available only at single sampling locations. Because precipitation isotopes vary with altitude and other factors, we did not interpolate individual station measurements across our study catchments, but instead relied on the monthly gridded precipitation isotope reanalysis database Piso.AI (Nelson *et al.*, 2021), which we averaged within the boundaries of each of our study catchments.




**Table 1: Data summary for study catchments, including original database ID, site name abbreviation (site code), river and gauge names, database name, latitude and longitude at catchment outlet, and number of samples for $\delta^{18}$O and $\delta^{2}$H in the publicly available streamflow isotope datasets: WISA (National Austrian Isotope database; n = 12), ISOT (National Swiss isotope database; n = 8) and CH-IRP (Staudinger *et al.*, 2020; n = 12).**


| org ID | Site code | River | Gauge | Data origin | long (deg.) | lat (deg.) | # $^{18}$O | # $^{2}$H |
|---|---|---|---|---|---|---|---|---|
| IO20000010 | DRA | Drau | Neubrücke | WISA | 14.46 | 32.36 | 162 | 162 |
| IO30000009 | DOH | Donau | Hainburg | WISA | 16.48 | 56.08 | 174 | 174 |
| IO30000014 | LEI | Leitha | Brodersdorf | WISA | 16.47 | 28.55 | 175 | 175 |
| IO30000015 | MAR | March | Angern | WISA | 16.48 | 49.25 | 156 | 156 |
| IO40000001 | DOE | Donau | Engelhartszell | WISA | 13.48 | 43.3 | 168 | 168 |
| IO40000012 | INS | Inn | Schärding | WISA | 13.48 | 25.26 | 172 | 172 |
| IO50000018 | SAL | Salzach | Salzburg | WISA | 13.47 | 4.44 | 172 | 172 |
| IO60000016 | MUS | Mur | Spielfeld | WISA | 15.46 | 37.42 | 175 | 175 |
| IO70000013 | INK | Inn | Kirchbichl | WISA | 12.47 | 4.3 | 176 | 176 |
| IO80000011 | ILL | Ill | Gisingen | WISA | 9.47 | 35.13 | 175 | 175 |
| IO80000017 | RHL | Rhine | Lustenau | WISA | 9.47 | 39.26 | 174 | 174 |
| IO90000005 | DOW | Donau | Wien-Nußdorf | WISA | 16.48 | 23.13 | 166 | 166 |
| NIO08 | RHW | Rhine | Weil | ISOT | 7.59 | 47.60 | 361 | 195 |
| NIO01 | AAB | Aare | Brienzwiler | ISOT | 8.09 | 46.75 | 347 | 346 |
| NIO07 | AAT | Aare | Thun | ISOT | 7.61 | 46.76 | 255 | 254 |
| NIO02 | AAR | Aare | Brugg | ISOT | 8.19 | 47.48 | 319 | 316 |
| NIO04 | RHP | Rhône | Porte du Scex | ISOT | 6.89 | 46.35 | 329 | 326 |
| NIO09 | RHC | Rhône | Chancy | ISOT | 5.97 | 46.15 | 102 | 102 |
| NIO05 | TIC | Ticino | Riazzino | ISOT | 8.91 | 46.16 | 287 | 285 |
| NIO06 | INE | Inn | S-chanf | ISOT | 10.00 | 46.62 | 226 | 226 |
| AAM | AAM | Aa | Mönchaltorf | CH_IRP | 8.47 | 47.19 | 95 | 95 |
| ALL | ALL | Allenbach | Adelboden | CH_IRP | 7.33 | 46.29 | 173 | 173 |
| ALP | ALP | Alp | Einsiedeln | CH_IRP | 8.44 | 47.09 | 319 | 319 |
| BIB | BIB | Biber | Biberbrugg | CH_IRP | 8.43 | 47.09 | 318 | 318 |
| DIS | DIS | Dischmabach | Davos | CH_IRP | 9.52 | 46.46 | 128 | 128 |
| ERG | ERG | Ergolz | Liestal | CH_IRP | 7.44 | 47.29 | 223 | 223 |
| ILF | ILF | Ilfis | Langnau | CH_IRP | 7.47 | 46.56 | 224 | 224 |
| LAN | LAN | Langeten | Huttwil | CH_IRP | 7.49 | 47.07 | 197 | 197 |
| MUR | MUR | Murg | Wängi | CH_IRP | 8.57 | 47.29 | 128 | 128 |
| SCH | SCH | Schaechen | Bürglen | CH_IRP | 8.39 | 46.52 | 181 | 181 |
| SEN | SEN | Sense | Thörishaus | CH_IRP | 7.21 | 46.53 | 198 | 198 |
| SIT | SIT | Sitter | Appenzell | CH_IRP | 9.24 | 47.19 | 185 | 185 |

## 2.3. Hydroclimatic variables

We assessed the relationship of isotopic signatures to 9 hydroclimatic variables (Table 2) in all 32 catchments. Daily catchment averaged precipitation from the gridded precipitation dataset E-OBS (version 20.0e) at 0.1-degree resolution (Cornes *et al.*, 2018) was used to calculate the mean annual, winter (November through April) and summer (May through



October) precipitation for each of the 32 catchments. The catchment boundaries were used to average mean monthly potential evapotranspiration (*PET*) across each catchment from the "Global Aridity Index and Potential Evapotranspiration

Climate Database v2" (Trabucco and Zomer, 2019); from these monthly averages, we also calculated annual, winter (November through April) and summer (May through October) *PET*. The discharge fraction ($q\ P^{-1}$) was calculated by dividing the total annual streamflow (in mm) by the total annual precipitation. The discharge that is exceeded 95% of the time ($q_{95}$ – in mm per day) was obtained from the streamflow duration curve by calculating the 5th percentile of all streamflow values. The use of $q$ instead of $Q$ indicates that the values are divided by area to obtain an area-normalized

discharge quantile (in mm per day rather than $m^3$ per second).

## 2.4. Physical catchment properties

We assessed the relationship between isotopic signatures to 9 physical catchment properties (Table 3) across all 32 study

catchments. From the catchment boundaries, the total catchment area was calculated with the "Tabulate Area" – tool of ArcMap 10.6. The mean, minimum, and maximum catchment elevation were calculated from the Copernicus EU-DEM v1.1 at 25 m resolution using the ArcMap 10.6 Zonal Statistics tool. The mean slope, fraction of slope below 10°, and fraction of slope above 40° were also calculated from the Copernicus EU-DEM v1.1. From the 0.5° resolution raster map GLiM ("Global Lithological Map" - Hartmann and Moosdorf, 2012), the fraction of area of potentially karstified carbonate

sedimentary rocks (GLiM class SC) and the fraction of area covered with unconsolidated debris deposits (GLiM class SU) were calculated. From the Copernicus CORINE land cover "CLC 2018" the fraction of area covered by forests was calculated, by combining the three CLC classes broad-leaved forest, coniferous forest, and mixed forest. All geodata mentioned above were extracted using the obtained catchment boundaries and the ArcMap 10.6 Zonal Statistics tool.

## 2.5. Calculations of young water fractions ($F_{yw}$) and new water fractions ($F_{new}$)

In seasonal climates, summer precipitation is isotopically heavier and winter precipitation is isotopically lighter, resulting in a seasonal cycle of precipitation isotopes. Kirchner (2016a; 2016b) showed that the young water fraction $F_{yw}$ (the fraction of streamflow that is younger than 2-3 months) can be estimated from the ratio of the seasonal amplitudes of sinusoidal fits to

precipitation and streamflow isotope time series. Fits that are robust against outliers can be obtained using iteratively re-weighted least squares (IRLS); an R script for this approach is available in the supplement of von Freyberg *et al.* (2018). We used results for the volume weighed $F_{yw}$, because estimates of $F_{yw}$ are more reliable when the sinusoidal isotope fits are volume-weighted by the precipitation and streamflow volumes, and when they are derived from longer time series that yield more stable amplitude estimates (Kirchner 2016a; 2016b; von Freyberg *et al.*, 2018; von Freyberg *et al.*, 2017). .




We also calculated new water fractions ($F_{new}$) using the ensemble hydrograph separation approach outlined in Kirchner (2019). A major advantage of $F_{new}$ over $F_{yw}$ is that the time scale over which water is considered "young" (2-3 months) depends on the shape of the catchment transit time distribution, which will typically be unknown, whereas the time scale over which water is "new" is directly linked to the sampling frequency of the isotope time series. That is, because our isotope

data is sampled at monthly resolution, $F_{new}$ estimates the fraction of streamwater that originated from precipitation in the one-month period since the previous streamwater sample was collected.

The ensemble hydrograph separation approach is based on correlations between isotopic fluctuations in streamflow and precipitation (and potentially also other endmembers). It estimates the average contribution of precipitation to streamflow

through correlations across an ensemble of precipitation and streamflow isotope samples. This makes it insensitive to unknown or unmeasured endmembers, and avoids the spurious results that can arise in traditional hydrograph separation when the "old water" and "new water" isotopic signatures overlap. While traditional hydrograph separation assesses how fractions of new and old water change over successive time steps (e.g., during an individual storm event), ensemble hydrograph separation can quantify the average fractions of new and old water over ensembles of non-successive time steps

reflecting different conditions (e.g., antecedent moisture). This is another major advantage of $F_{new}$ over $F_{yw}$, and a main goal of this study is to calculate new water fractions across all 32 Alpine catchments for the entire dataset and for subsets of the data reflecting different catchment conditions. Here, we report volume-weighted new water fractions of streamflow ($^{Q}F^{*}_{new}$ in the notation of Kirchner (2019)) calculated using the R script provided by Kirchner and Knapp (2020a) and Kirchner and Knapp (2020b).


To assess $F_{new}$ for the driest and wettest half of the dataset, we split our data based on monthly precipitation totals recorded prior to sampling. To assess seasonal differences, we also split the data into the winter (November through April) and summer (May through October) halves of the year. For each of these four subsets of data we then calculated $F_{new}$. We also calculated the fraction of new water as a function of incoming precipitation (as described in Section 3.5 of Kirchner 2019;

Section 5.4 of Knapp *et al.*, 2019). We expect that higher values of $F_{new}$ will often be associated with higher precipitation totals in the month immediately preceding sampling.

Previous studies have indicated that the (volume weighted) fractions of new and young water are related to physical catchment properties (e.g., to mean catchment elevation). Thus, for all 32 catchments we calculated the following seven

topographic properties: total catchment area, mean catchment elevation, elevation difference (calculated from maximum elevation – minimum elevation), mean catchment slope, the fraction of slope below 10°, and the fraction of slope above 40°. To assess the effect of lithology, we calculated the fraction of (potentially karstified) carbonate sedimentary rocks and the fraction of unconsolidated debris deposits for all catchments. Because plants, such as trees, affect the hydrological cycle by increasing water losses through transpiration, we also calculated the fraction of forest cover for each catchment.




## 2.6. Statistical measures for data analysis

Isotope data are presented in the delta notation in per mil (‰) of ¹⁸O relative to V-SMOW (Vienna Standard Mean Ocean Water) throughout the paper (the respective ²H plots can be found in the supplement). Overall, both $\delta^{18}O$ and $\delta^2H$ should

yield similar results if non-equilibrium isotope fractionation processes due to evaporation can be neglected (Craig and Gordon, 1965). Some data are presented in boxplots, in which the horizontal line indicates the median, the box represents the interquartile range, and the whiskers extend to 1.5 times the interquartile range from the first and third quartiles (or to the maximum and minimum values). The dots indicate outliers. Spearman rank correlations were calculated to obtain the correlation coefficient $r_S$ and its associated $p$-value. Differences between groups of samples were tested using Wilcoxon

Signed-Rank tests. Results are presented as statistically significant when $p < 0.05$.

## 3.  Results

### 3.1. Hydroclimatic variables and catchment characteristics

Among the 32 study sites, average annual precipitation ranges from 611 mm to 1741 mm (mean = 1244 mm), winter

precipitation ranges from 211 mm to 748 mm (mean = 534 mm) and summer precipitation ranges from 400 mm to 994 mm (mean = 710 mm).  Mean annual *PET* ranges from 669 mm to 912 mm (mean = 825 mm), winter *PET* ranges from 193 mm to 269 mm (mean = 240 mm) and summer *PET* ranges from 496 mm to 657 mm (mean = 586 mm). The ratio of average discharge to average precipitation ($q\,P^{-1}$) ranges from 0.10 to 0.78 (mean = 0.38); the $q_{95}$ ranges from 0.09 mm day$^{-1}$ to 1.54 mm day$^{-1}$ (mean = 0.83 mm day$^{-1}$).


**Table 2: Averages of major hydro-climatic variables across the 32 Alpine catchments: mean annual precipitation, winter precipitation, summer precipitation, mean annual potential evapotranspiration, winter potential evapotranspiration, summer potential evapotranspiration (all in mm), the ratio of annual streamflow to annual**

**precipitation (%) and the discharge reached or exceeded 95% of the time (in mm day$^{-1}$).**

| Site code | mean annual *P* | winter *P* | summer *P* | annual *PET* | winter *PET* | summer *PET* | $q\,P^{-1}$ | $q_{95}$ |
|---|---|---|---|---|---|---|---|---|
| [-] | mm | mm | mm | mm | mm | mm | % | mm day$^{-1}$ |
| DRA | 1368 | 534 | 834 | 769 | 215 | 554 | 0.30 | 0.97 |
| DOH | 995 | 401 | 594 | 816 | 225 | 591 | 0.29 | 0.80 |
| LEI | 887 | 333 | 554 | 912 | 259 | 653 | 0.11 | 0.19 |
| MAR | 611 | 211 | 400 | 891 | 234 | 657 | 0.10 | 0.09 |
| DOE | 1008 | 409 | 599 | 809 | 223 | 586 | 0.28 | 0.81 |

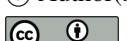



| | | | | | | | | |
|---|---|---|---|---|---|---|---|---|
| INS | 1176 | 466 | 710 | 785 | 224 | 561 | 0.40 | 1.20 |
| SAL | 1364 | 528 | 836 | 745 | 216 | 529 | 0.53 | 1.54 |
| MUS | 1213 | 448 | 764 | 799 | 226 | 573 | 0.21 | 0.59 |
| INK | 1094 | 443 | 651 | 746 | 215 | 531 | 0.46 | 1.07 |
| ILL | 1374 | 572 | 802 | 809 | 245 | 564 | 0.59 | 1.46 |
| RHL | 1417 | 586 | 831 | 797 | 240 | 557 | 0.40 | 1.16 |
| DOW | 1003 | 405 | 599 | 812 | 224 | 588 | 0.29 | 0.80 |
| RHW | 1290 | 569 | 721 | 850 | 246 | 604 | 0.34 | 1.24 |
| AAB | 1271 | 586 | 685 | 774 | 231 | 543 | 0.78 | 1.37 |
| AAT | 1201 | 580 | 620 | 822 | 250 | 572 | 0.58 | 1.19 |
| AAR | 1210 | 564 | 646 | 878 | 254 | 624 | 0.35 | 1.06 |
| RHP | 1086 | 564 | 523 | 813 | 244 | 569 | 0.51 | 1.26 |
| RHC | 1179 | 604 | 576 | 875 | 260 | 615 | 0.45 | 0.97 |
| TIC | 1480 | 585 | 895 | 856 | 268 | 588 | 0.44 | 1.08 |
| INE | 974 | 408 | 566 | 689 | 193 | 496 | 0.61 | 0.60 |
| AAM | 1208 | 487 | 721 | 888 | 244 | 644 | 0.27 | 0.23 |
| ALL | 1251 | 661 | 590 | 853 | 269 | 584 | 0.56 | 1.29 |
| ALP | 1625 | 674 | 951 | 859 | 258 | 601 | 0.44 | 0.61 |
| BIB | 1543 | 637 | 905 | 868 | 260 | 608 | 0.36 | 0.35 |
| DIS | 1211 | 522 | 690 | 690 | 194 | 497 | 0.52 | 0.72 |
| ERG | 1286 | 613 | 673 | 872 | 246 | 626 | 0.17 | 0.11 |
| ILF | 1364 | 621 | 743 | 868 | 258 | 610 | 0.32 | 0.53 |
| LAN | 1322 | 599 | 723 | 886 | 250 | 637 | 0.23 | 0.73 |
| MUR | 1167 | 471 | 696 | 858 | 239 | 619 | 0.30 | 0.37 |
| SCH | 1741 | 748 | 994 | 805 | 247 | 557 | 0.46 | 1.10 |
| SEN | 1240 | 581 | 659 | 884 | 262 | 622 | 0.30 | 0.52 |
| SIT | 1651 | 685 | 967 | 831 | 250 | 581 | 0.35 | 0.54 |

The catchments range in size from 29 km$^2$ to 103'946 km$^2$ (mean = 13'923 km$^2$, median = 1562 km$^2$). The mean catchment elevation varies from 379 m a.s.l. to 2472 m a.s.l. (mean = 1310 m a.s.l.), and the elevation difference varies from 403 m to 4454 m (mean = 2477 m). The mean catchment slope ranges from 4.6° to 50.9° (mean = 20.1°), the fraction of catchment area with slope < 10° ranges from 9.1 % to 93.2 % (mean = 65 %), and the fraction of catchment area with slope > 40° ranges from 0 to 18.5 % (mean = 6.1 %). The fraction of catchment area covered by carbonate sedimentary rocks ranges from 0 to 80.1 % (mean = 27.4 %), the fraction of area covered by unconsolidated rocks ranges from 0 to 56.1 % (mean = 17.5 %). The fraction of catchment area covered by forests ranges from 3.1 % to 60.3 % (mean = 32.9 %).

**Table 3: Catchment averages of physical catchment properties across the 32 Alpine catchments: total catchment area, mean catchment elevation, elevation difference (max-min), mean catchment slope, fraction of area with slope < 10°, fraction of area with slope > 40°, fraction of area with potentially karstified carbonate sedimentary rocks, the fraction of area with unconsolidated debris deposits, and the fraction of area covered by forests.**





| Site code | area | mean elevation | elevation difference | mean slope | f slope < 10° | f slope > 40° | f karstified | f debris | f forest |
|---|---|---|---|---|---|---|---|---|---|
| [-] | km² | m asl. | m | ° | % | % | % | % | % |
| DRA | 10398 | 1389.8 | 3336.9 | 20.8 | 78.7 | 5.8 | 23.0 | 0.7 | 52.0 |
| DOH | 103946 | 796.0 | 3843.6 | 10.2 | 33.6 | 2.4 | 25.6 | 19.2 | 37.1 |
| LEI | 1588 | 699.1 | 1864.7 | 12.9 | 54.5 | 1.2 | 40.3 | 0.0 | 60.3 |
| MAR | 25616 | 379.3 | 1051.5 | 4.6 | 11.1 | 0.0 | 2.1 | 27.0 | 30.4 |
| DOE | 77107 | 838.7 | 3701.1 | 9.7 | 30.4 | 2.6 | 24.1 | 25.7 | 34.4 |
| INS | 24232 | 1321.7 | 3676.2 | 18.0 | 63.0 | 6.2 | 27.9 | 17.1 | 35.9 |
| SAL | 3910 | 1513.8 | 3181.3 | 24.0 | 87.0 | 8.5 | 32.7 | 0.0 | 43.3 |
| MUS | 9575 | 1074.2 | 2770.5 | 17.3 | 71.8 | 1.8 | 18.2 | 0.0 | 59.1 |
| INK | 9304 | 1941.2 | 3494.2 | 25.2 | 88.9 | 10.0 | 26.9 | 3.6 | 30.0 |
| ILL | 1282 | 1608.2 | 2773.5 | 25.1 | 87.9 | 10.3 | 42.2 | 0.6 | 36.0 |
| RHL | 6500 | 1736.6 | 3150.8 | 23.4 | 85.3 | 8.5 | 40.8 | 18.3 | 29.3 |
| DOW | 101803 | 806.4 | 3829.5 | 10.3 | 33.7 | 2.5 | 25.7 | 19.6 | 37.0 |
| RHW | 36435 | 1049.9 | 3899.2 | 14.1 | 51.7 | 4.3 | 28.2 | 31.4 | 31.7 |
| AAB | 587 | 2101.3 | 3562.1 | 27.5 | 88.7 | 17.5 | 23.8 | 7.0 | 15.5 |
| AAT | 2521 | 1739.9 | 3580.0 | 24.7 | 84.3 | 14.0 | 51.5 | 15.8 | 23.4 |
| AAR | 11584 | 1006.7 | 3794.8 | 12.8 | 47.0 | 3.9 | 29.6 | 35.3 | 30.0 |
| RHP | 5307 | 2096.7 | 4144.2 | 25.5 | 87.8 | 12.2 | 21.5 | 17.8 | 23.5 |
| RHC | 10309 | 1564.9 | 4454.2 | 19.7 | 68.6 | 8.5 | 25.1 | 22.2 | 27.8 |
| TIC | 1562 | 1651.7 | 3140.0 | 28.7 | 91.2 | 18.5 | 9.2 | 12.0 | 44.1 |
| INE | 625 | 2472.4 | 2327.3 | 50.9 | 86.6 | 8.7 | 13.2 | 17.3 | 11.3 |
| AAM | 49 | 522.3 | 402.5 | 6.1 | 9.1 | 0.0 | 0.0 | 23.8 | 15.8 |
| ALL | 29 | 1866.3 | 1390.2 | 28.8 | 89.7 | 8.1 | 80.1 | 17.7 | 19.7 |
| ALP | 46 | 1160.4 | 854.7 | 19.4 | 70.8 | 0.7 | 48.3 | 30.6 | 50.4 |
| BIB | 32 | 1009.4 | 637.8 | 14.9 | 53.1 | 0.0 | 0.8 | 56.1 | 42.9 |
| DIS | 43 | 2372.1 | 1401.7 | 28.7 | 92.7 | 6.3 | 0.0 | 18.6 | 3.1 |
| ERG | 261 | 588.8 | 855.0 | 15.0 | 56.0 | 0.1 | 78.0 | 4.9 | 43.9 |
| ILF | 188 | 1050.1 | 1361.8 | 25.0 | 78.1 | 1.1 | 5.8 | 7.4 | 53.8 |
| LAN | 60 | 760.9 | 450.4 | 12.2 | 31.6 | 0.0 | 0.0 | 21.5 | 15.1 |
| MUR | 77 | 652.3 | 538.6 | 12.5 | 36.6 | 0.0 | 0.0 | 37.9 | 33.3 |
| SCH | 108 | 1738.7 | 2618.1 | 31.1 | 93.2 | 16.4 | 36.0 | 17.9 | 21.2 |
| SEN | 351 | 1079.8 | 1566.6 | 18.1 | 61.3 | 1.5 | 31.5 | 24.3 | 35.0 |
| SIT | 88 | 1316.9 | 1613.1 | 25.9 | 77.0 | 13.4 | 63.0 | 7.2 | 27.8 |

**3.2. Isotopic variation of precipitation and streamflow across Alpine catchments**

The seasonal variations of isotope ratios in precipitation and streamflow are expected to be related to the size of the catchment (i.e., more damped streamflow isotope ratios in larger catchments) and the mean catchment elevation (i.e., precipitation isotope ratios are lighter at higher elevations). The relationships of precipitation and streamflow δ¹⁸O signatures to catchment area and elevation are shown in Figure 2. Overall, the amplitudes of the seasonal isotope signatures are damped from precipitation to streamflow. While there is only a weak trend between the isotope ratios and catchment area,

precipitation and streamflow isotope ratios are lighter in higher-elevation catchments as expected.



The Spearman rank correlation coefficients ($r_S$) between catchment area and the median $\delta^{18}O$ for precipitation and streamflow were 0.28 and 0.02, respectively (not significant). The range of variation (max-min) of $\delta^{18}O$ was inversely correlated with catchment area; $r_S$ = -0.71 ($p$ = < 0.05) and -0.45 ($p$ = < 0.05) for precipitation and streamflow, respectively.

This suggests that the median of $\delta^{18}O$ precipitation and streamflow isotopes only slightly increases with catchment size, whereas precipitation and streamflow isotopes are significantly less variable over time in larger catchments. Median $\delta^{18}O$ decreased with increasing catchment elevation; $r_S$ = -0.85 ($p$ = < 0.05) and -0.90 ($p$ = < 0.05) for precipitation and streamflow, respectively. Similar results were obtained for $\delta^{2}H$; see Supplementary Material (Figure S1).

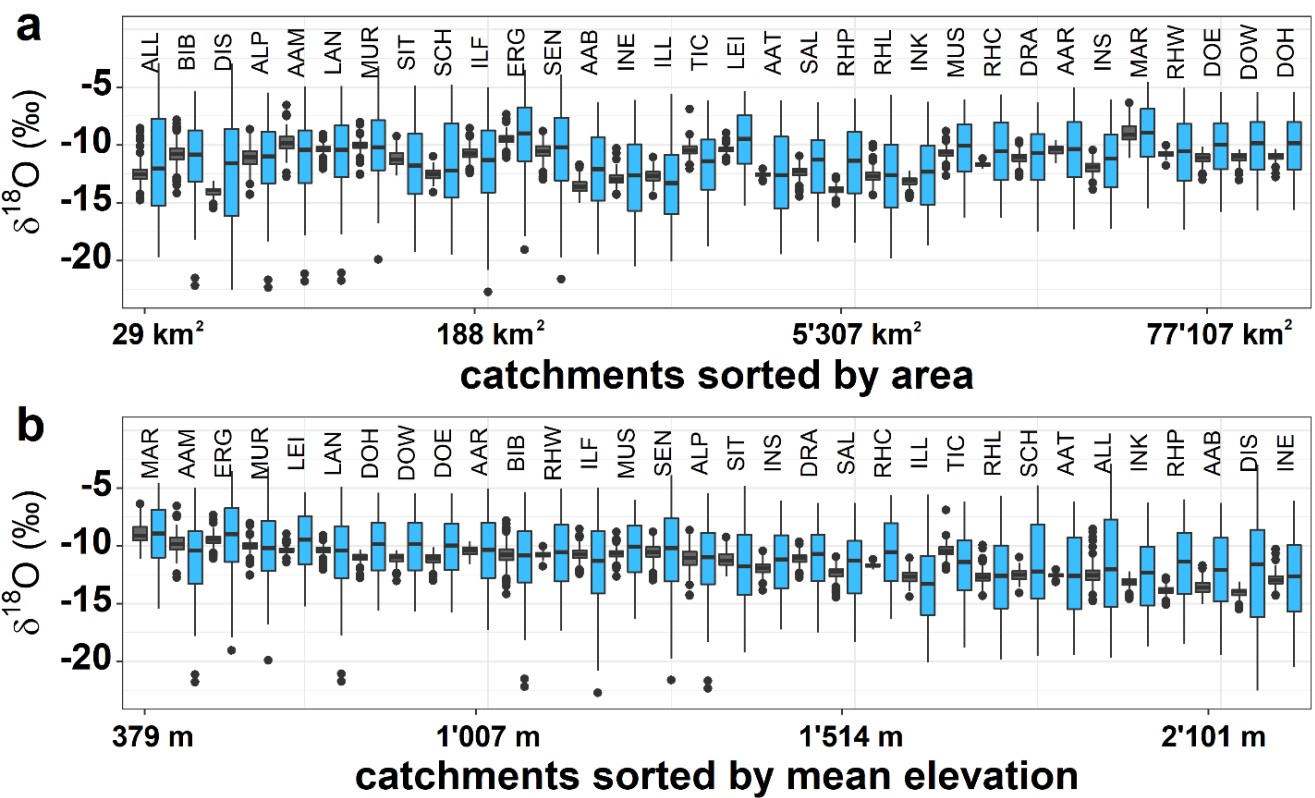


**Figure 2: Boxplots of the $\delta^{18}O$ isotopic composition of precipitation (light blue) and streamflow (dark grey) across all 32 Alpine catchments sorted by catchment area from small to large (a) and mean catchment elevation from low to high (b).**



### 3.3. New ($F_{new}$) and young ($F_{yw}$) water fractions across Alpine catchments

In Figure 3, the fractions of new water ($F_{new}$) and young water ($F_{yw}$) are shown. The catchments are sorted by mean catchment elevation from MAR (March at Angern, 379 m a.s.l.) to INE (Inn at S-canf, 2472 m a.s.l.). Both $F_{new}$ and $F_{yw}$ tended to be smaller at higher mean catchment elevations ($r_S$ = -0.37 and -0.32). However, the study catchments encompass different precipitation and discharge regimes: MAR to BIB are rainfall-dominated (precipitation falls almost exclusively as rain), RHW to DRA are hybrid (both rain and snow can contribute significantly to winter precipitation), and SAL to INE are snow-dominated (most winter precipitation falls as snow). Both $F_{new}$ and $F_{yw}$ differed significantly between rainfall-dominated and snow-dominated catchments as well as between hybrid and snow-dominated catchments ($p < 0.05$). Mean $F_{new}$ varied from 9.2 % in rainfall-dominated catchments and 9.6 % in hybrid catchments to 3.5 % in snow-dominated catchments, and mean $F_{yw}$ varied from 17.6 % in rainfall-dominated, 16.6 % in hybrid catchments to 10.1 % in snow-dominated catchments. $F_{new}$ and $F_{yw}$ were strongly correlated ($R = 0.88$), with $F_{yw}$ systematically exceeding $F_{new}$ (Figure 3c). This is expected because $F_{yw}$ expresses the fraction of water younger than 2-3 months, which should always be greater than the fraction of water younger than one month (i.e., $F_{new}$ estimated from monthly data).





**Figure 3: New water fractions (a) and young water fractions (b) for all catchments sorted by elevation.** $F_{new}$ **and** $F_{yw}$ **are smaller in catchments of higher mean elevation.** $F_{new}$ **and** $F_{yw}$ **are strongly correlated, with** $F_{yw}$ **being systematically higher as indicated by the points lying above the 1:1 line (c).**





### 3.4. New water fractions ($F_{new}$) across Alpine catchments for different subsets of data

In the following section, the potential of the $F_{new}$ analysis is further explored, as $F_{new}$ can be calculated for different subsets of the data; for example, for different antecedent conditions or different seasons. For most catchments, $F_{new}$ was larger in the wettest (highest-precipitation) 50% of all months than in the driest half (mean $F_{new}$ = 9.3 % and 3.3 %, respectively; $p <$ 0.05). $F_{new}$ also tended to be larger in summer than in winter (mean $F_{new}$ = 12.7 % and 8.9 %, respectively; $p < 0.05$). This is not surprising, because across most of the Alps, precipitation is typically higher in the summer than the winter (mean across all catchments of our study 710 mm in summer and 534 mm in winter; only RHP and ALL receive more precipitation in winter than in summer).

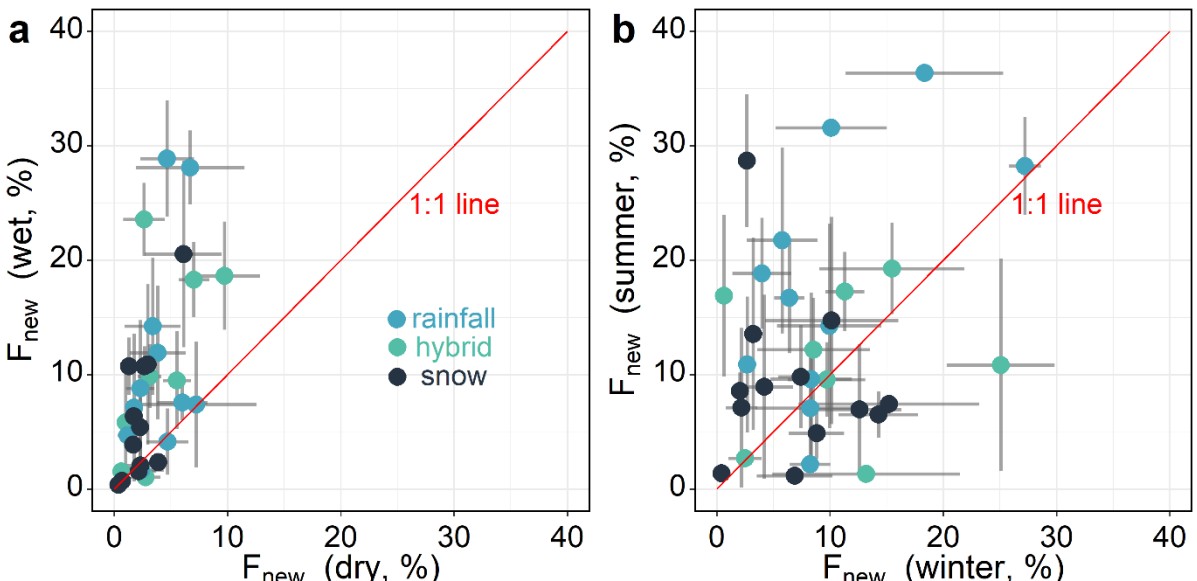

**Figure 4: New water fractions for the driest and wettest half of the dataset (a), and for the winter and summer half of the dataset (b). The colours indicate the different precipitation regimes (rainfall, hybrid, snow). New water fractions tend to be higher in wet periods with few exceptions. Summer new water fractions tend to be higher in most catchments.**

We also calculated the fraction of new water for different ranges of monthly precipitation. It is expected that $F_{new}$ is larger for months with more incoming precipitation. As shown in Figure 5, 18 of our 32 study catchments show increases in $F_{new}$ above a certain threshold in monthly precipitation rates, i.e. roughly 70 mm for MAR, roughly 110 mm for AAM, 175 mm for MUS, DOE, DOH, DOW, ERG, MUR, RHL, BIB and LEI, roughly 200 mm for SEN, DOW, LAN, INK, SIT and INS, and roughly 225 mm for SAL. For these catchments, it is evident that - after the threshold precipitation inputs are reached -



the more recent precipitation fell, the more recent precipitation can be found in streamflow, as indicated by increasing $F_{new}$. For 14 out of 32 catchments, more incoming precipitation does not raise $F_{new}$ above ~10% (Figure 5b), suggesting catchment storage is large enough to damp tracer fluctuations, even under high monthly precipitation rates. These different responses to incoming precipitation are associated with differences in catchment elevation and slope. The eighteen catchments in which $F_{new}$ increased substantially above a precipitation threshold (Figure 5a) have a mean elevation of 1068 m a.s.l. and a mean slope of 16.1°, whereas the 14 catchments in which $F_{new}$ remained small (Figure 5b) are, on average, both higher (mean elevation of 1584 m a.s.l.) and steeper (mean slope of 24.3°); both differences are statistically significant ($p < 0.05$).

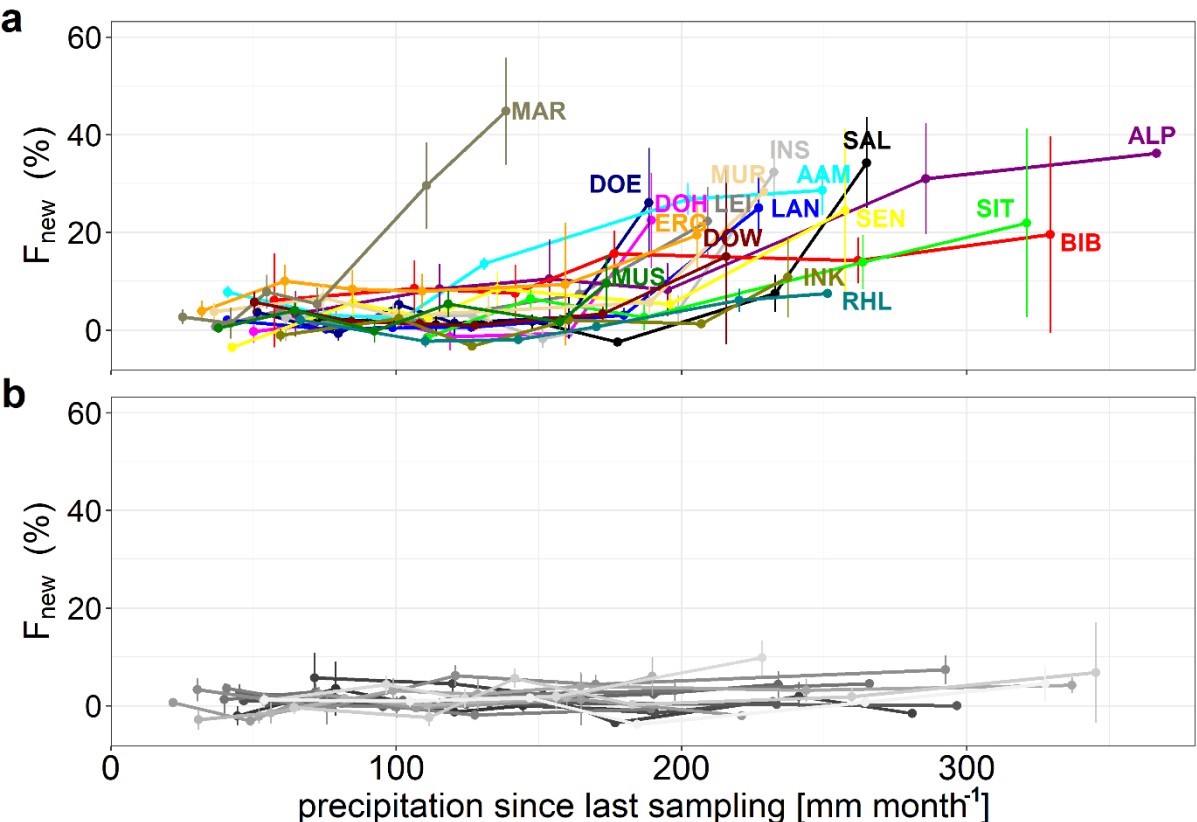

**Figure 5: The volume weighted fraction of new water ($F_{new}$) as a function of monthly precipitation totals during the month immediately preceding the sampling date. 18 out of the 32 catchments, as shown in (a), exhibit a clear increase in $F_{new}$ at higher monthly precipitation totals. In these cases, $F_{new}$ increases at monthly precipitation rates exceeding ~175 mm month$^{-1}$. For 14 out of 32 catchments, $F_{new}$ remains below 10% even at the highest monthly precipitation totals (b).**





### 3.5. Downstream propagation of $F_{new}$ in Danube and Rhine

New water fractions $F_{new}$ were mapped across 7 sub-catchments of the Danube river basin (Figure 6 a & c). Two headwater
basins of the Inn have exceptionally small $F_{new}$ values (0.7 % at INE and 2.3% at INK), potentially due to snowpack storage
in these high-elevation catchments. It should also be noted that INE is sampled directly below several large lakes (St. Moritz,
Engadin, Switzerland), and thus the damped isotope signal probably reflects mixing and storage within those lakes. The
remaining 5 Danube catchments exhibit a weak declining trend in $F_{new}$ with increasing catchment area, which nonetheless
results in a perfect rank correlation ($r_S$=-1.0) due to the small sample size (if all 7 sub-catchments are considered together,
the rank correlation is non-significant).

The new water fractions $F_{new}$ were also mapped across the 18 subcatchments of the Rhine river basin (Figure 6 b & d). The
smaller headwater catchments tend to have larger $F_{new}$ (i.e., $F_{new}$ exceeds 8% for BIB, ALP, AAM, SIT, and SEN),
suggesting that these headwater streams contain relatively large proportions of recent precipitation. These five headwater
catchments are at low to intermediate mean elevations (522-1317 m a.s.l.). Across the Rhine basin, smaller catchments tend
to have higher $F_{new}$ ($r_S$ = -0.67, p<0.05; Figure 6d).









**Figure 6: New water fractions ($F_{new}$) mapped across the subcatchments of the Danube (a) and Rhine (b) basins (n=7 and n=18, respectively), and $F_{new}$ as a function of catchment area for the Danube (c) and Rhine (d) subcatchments. The Inn headwater catchments in the southeast of the Danube basin (INE and INK) have small $F_{new}$. For the other**
**subcatchments in the Danube and Rhine basins, the expected decrease in $F_{new}$ with increasing catchment size is evident. Note that the x-axis in subpanel (d) is logarithmic. The hillshades in the maps are based on the EU-DEM v1.1. available through funding by the European Union.**

### 3.6. Relationships between $F_{new}$ and hydroclimatic variables

Previous studies indicated that the (volume weighted) fractions of new water are related to hydroclimatic variables, e.g., catchments have larger $F_{new}$ following wet antecedent conditions (Knapp *et al.*, 2019). Thus, we analyzed the relationships between $F_{new}$ and a range of climate variables (annual, summer, and winter precipitation sums, and annual, summer and winter evapotranspiration sums) as well as the ratio of total discharge to total precipitation ($q\,P^{-1}$), and the discharge that is
reached or exceeded 95% of the year ($q_{95}$). As Figure 7 shows, $F_{new}$ values were only weakly related to catchment annual, winter, and summer precipitation. Thus, mean annual or seasonal precipitation is a poor predictor of $F_{new}$, despite the fact that $F_{new}$ does tend to be higher in wetter months (Figure 4). $F_{new}$ was more strongly correlated to the amount of annual *PET* and summer *PET* ($r_S = 0.41$ and $0.40$, respectively). $F_{new}$ was also negatively correlated with the $q\,P^{-1}$ ratio ($r_S = -0.36$, $p < 0.05$) or, positively correlated with the fraction $(P-Q)/P$ of precipitation that was evaporated and transpired ($r_S = 0.36$, $p <$
$0.05$). Thus, site-to-site variations in $F_{new}$ were positively correlated with both potential and actual evapotranspiration. $F_{new}$ was also inversely correlated with $q_{95}$ ($r_S = -0.52$ at $p < 0.05$), suggesting higher $F_{new}$ in catchments with smaller baseflow.

As new and young water fractions are strongly correlated with each other ($r_S = 0.88$, $p < 0.05$), $F_{yw}$ exhibited similar correlations with hydroclimatic variables as those that were found for $F_{new}$ (see Supplementary Material Figure S2).





**Figure 7: Relationships between volume-weighted new water fractions ($F_{new}$) and (a) annual precipitation, (b) summer (May through October) precipitation, (c) winter (November through April) precipitation, (d) annual potential evapotranspiration, (e) summer (May through October) potential evapotranspiration, (f) winter (November through April) potential evapotranspiration, (g) the fraction of annual discharge in relation to annual precipitation ($q P^{-1}$), and (h) $q_{95}$, the discharge reached or exceeded 95% of the year. While $F_{new}$ is not strongly related to precipitation, $F_{new}$ is related to $PET$ and the streamflow variables.**



### 3.7. Relations of $F_{new}$ to physical catchment properties


Previous studies indicate that the fractions of new (and young) water may be related to physical catchment properties. Therefore, across all 32 study catchments we compared the isotopically inferred $F_{new}$ values with the catchment area, mean catchment elevation, elevation difference, the ratio of elevation difference to total catchment area, as well as mean catchment slope and the fraction of slope below 10° and above 40°. To assess the effect of lithology, the fraction of (potentially

karstified) carbonate sedimentary rocks and the fraction of unconsolidated debris deposits for all catchments were calculated. To assess the possible role of transpiration, the fraction of forest cover for each catchment was also calculated.

Figure 8 shows a systematic assessment of the relation of $F_{new}$ and physical catchment properties. $F_{new}$ was inversely related to catchment area, mean elevation, and elevation difference ($r_S$ = -0.38, -0.37, and -0.59, respectively, all $p < 0.05$). Thus,

$F_{new}$ was smaller in larger catchments, higher catchments, and catchments with greater total relief. However, these three drivers are themselves strongly correlated with one another, so it is unclear which one could be considered as the primary control on $F_{new}$. Although the average catchment slope was not a good predictor for $F_{new}$ ($r_S$ = -0.20, not significant), there was a much stronger correlation between $F_{new}$ and the fraction of slope larger than 40° ($r_S$ = -0.48, $p < 0.05$). This suggests that catchments with larger fractions of steep slopes have smaller $F_{new}$, but note that the fraction of slope larger 40° was also

correlated to mean catchment elevation ($r_S$ = 0.84). The fractions of potentially karstified carbonate sedimentary rocks and unconsolidated debris deposits were not related to $F_{new}$. Statistically significant correlations between the fraction of catchment area covered by forests and $F_{new}$ ($r_S$ = 0.36 at $p < 0.05$) suggest a possible role for forest vegetation in shaping flowpaths and thus water ages.

Young water fractions exhibited similar relationships to catchment properties (see Supplementary Material Figure S3).as those observed for $F_{new}$.







**Figure 8: Relation of volume weighted new water fractions ($F_{new}$) and (a) catchment area, (b) mean elevation, (c) elevation difference, (d) mean slope, (e) fraction of slope shallower than 10°, (f) fraction of slope steeper than 40°, (g) fraction of the catchment consisting of karstified rocks, (h) fraction of the catchment covered by unconsolidated rocks, and (i) and fraction of the catchment covered by forests. $F_{new}$ is strongly related to catchment area, elevation difference, fraction of slope steeper than 40°, and fraction of the catchment covered by forest.**



## 4. Discussion

### 4.1. New ($F_{new}$) and young ($F_{yw}$) water fractions across Alpine catchments

Across our 32 study catchments, $F_{new}$ and $F_{yw}$ decreased with increasing catchment elevation (Figures 3, 8b, and S4). This is consistent with previous findings of Ceperley *et al.* (2020), who observed a decrease in young water fractions above an elevation of 1500 m asl. across Swiss Alpine catchments. Conversely, von Freyberg *et al.* (2018) found weak positive correlations with catchment elevation for a subset of our study catchments (12 of our catchments overlap with their set of 22 catchments). The relative importance of rain versus snow had a clear effect on $F_{new}$ and $F_{yw}$, which were lower in hybrid catchments than in rainfall-dominated catchments, and lower still in snow-dominated catchments (Figure 3). Conversely, Gentile *et al.*, (2023) found the highest $F_{yw}$ in hybrid catchments, and similarly low $F_{yw}$ in rainfall- and snow-dominated catchments. The reason for discrepancy might be that our dataset also covers much larger catchments (catchment areas in our dataset ranged from 29 km$^2$ to 103'946 km$^2$, versus 0.14 km$^2$ to 351 km$^2$ in Gentile *et al.* (2023).

$F_{new}$ was larger in wet periods than dry periods (with means of 9.3 and 3.3 %, respectively), and larger in the summer half of the year than in the winter half of the year (means of 12.7 and 8.9 %, respectively). Similar analyses were performed on the Plynlimon dataset by Knapp *et al.* (2019), who found that wetter antecedent conditions led to higher $F_{new}$. At Plynlimon, $F_{new}$ was smaller in summer (which is the drier season there), whereas in our 32 Alpine catchments, $F_{new}$ was smaller in winter (when precipitation rates are lower, and more precipitation falls as snow). Knapp *et al.* (2019) observed strong increases in 7-hourly and weekly $F_{new}$ above a precipitation threshold of roughly 5 mm day$^{-1}$, whereas we found that only 8 of 32 of our catchments showed strong increases in monthly $F_{new}$ above a comparable precipitation threshold of roughly 175 mm month$^{-1}$. Readers should note that the Plynlimon subcatchments studied by Knapp *et al.* (2019) are substantially smaller (< 3.6 km$^2$) than ours (> 29 km$^2$), and the temporal resolution of their data was much higher (7h to weekly) than ours (monthly).

Overall, $F_{new}$ tended to decrease downstream, from smaller headwaters to larger river basins (Section 3.5), which is consistent with the larger mean residence times that are typically estimated in larger catchments (DeWalle *et al.*, 1995; Soulsby *et al.*, 2000). Another important factor when moving from the headwaters downstream is the impact of water storage in lakes and reservoirs, as well as the potential effects of anthropogenic flow regulation. While the CH-IRP headwater catchments (Staudinger *et al.*, 2020) were carefully selected to avoid major impoundments or diversions, such complications are unavoidable in the larger basins contained in our dataset. Within the Danube basin, for example, INE has several large lakes just upstream from the sampling location, and within the Rhine basin, AAT (immediately below the lake of Thun) and RHW (approximately 100 km downstream from Lake Constance) are significantly impacted by lakes. Big lakes can substantially dampen $F_{new}$ by storing months or years of flow. Such storage yields exceptionally low $F_{new}$ values (for example, $F_{new}$ is 0.7, 0.4, and 0.6 % at INE, AAT, and RHW, respectively). $F_{new}$ can also be altered by dams and their





accompanying reservoirs; for example, multiple large dams lie upstream of DOE, DOW, and DOH, which have $F_{new}$ values

of 6.7, 5.1, and 4.1 % (for comparison, the mean of all rainfall-dominated catchments is 9.6%).

## 4.2. Conceptualization of physical drivers of $F_{new}$ and $F_{yw}$ across the Alps

When looking at the effects of hydroclimatic variables and physical catchment properties on $F_{new}$, single-variable

correlations with $F_{new}$ should be interpreted with caution, because they may be confounded by cross-correlations between many potential drivers (Figure **9**). For example, annual and summer $PET$, discharge fraction, $q_{95}$, mean slope, fraction of catchment area with slope > 40°, and fraction of catchment area with slope < 10° are all strongly related to mean catchment elevation. Thus, it remains unclear which of these variables may be a first-order control on new water fractions.



**Figure 9: Spearman rank correlations between all the selected hydroclimatic variables and physical catchment characteristics across the 32 study catchments. Red colours indicate strong to intermediate positive correlations; blue colours indicate strong to intermediate inverse correlations.**


Overall, we found that high fractions of new water ($F_{new}$) were more likely in small catchments, at low elevations, with small total relief and larger forest cover, and following months with high precipitation, whereas low $F_{new}$ values were more likely in large catchments, at high elevations, with large total relief, and following months with low precipitation (Figure **10**).


Von Freyberg *et al.* (2018) found significant correlations between $F_{yw}$ and monthly precipitation. Conversely, site-to-site differences in average $F_{new}$ and $F_{yw}$ were only weakly correlated with precipitation (annual, summer, and winter) across our 32 sites. Nevertheless, higher precipitation in the month preceding sampling typically led to larger $F_{new}$, which is in line with previous findings in Knapp *et al.* (2019). Higher $F_{new}$ values were typically found during the summer period, which receives

more precipitation than the winter period (see Section 3.3).

Across the Alps, elevation is correlated with $F_{new}$, but also with many other variables (e.g., temperature, precipitation, PET, and slope) that arguably should have stronger mechanistic connections with new water fractions. Site-to-site differences in $F_{new}$ were positively correlated with both annual *PET* and summer *PET*, and with long-term actual evapotranspiration,

estimated as ET/P=(P-Q)/P (see Section 3.6). This is somewhat counterintuitive, as one might expect that if more precipitation leaves the catchment via evapotranspiration, the smaller resulting discharge flux would imply longer retention times in the catchment and thus smaller new water fractions.

Across our 32 sites, $F_{new}$ exhibited a strong inverse correlation with the low-flow variable $q_{95}$ were found, suggesting higher

$F_{new}$ in catchments with smaller low flows. This is consistent with the findings of von Freyberg *et al.* (2018) showing that the quick-flow index (*QFI*) is positively correlated with $F_{yw}$. Although *QFI* and $q_{95}$ are not the same, they are systematically related, as high $q_{95}$ is more likely in catchments with low *QFI*.

We observed modest (but statistically significant) negative correlations between $F_{new}$ and catchment area, and von Freyberg

*et al.* (2018) also found weak (but non-significant) correlations between catchment area and $F_{yw}$. However, these relationships might be confounded by differences in the dominant precipitation regime; the correlation between $F_{new}$ and catchment area is stronger among our hybrid catchments ($R$ = -0.69, $p$ < 0.05) than among our rainfall-dominated or snow-dominated catchments (-0.27 and -0.39, respectively). Thus, $F_{new}$ (and $F_{yw}$) appear to be inversely related to catchment area,



but not so much in catchments with seasonal snow cover or glaciers (Gentile *et al.*, 2023; Ceperley *et al.*, 2020), which
generally have small $F_{new}$ and $F_{yw}$ values independent of their catchment area.

We found only a weak and non-significant inverse correlation between $F_{new}$ and mean catchment slope ($r_S$=-0.18), but strong inverse correlations between $F_{new}$ and both total relief (elevation difference; $r_S$=-0.59) and the fraction of catchment area with slopes steeper than 40° ($r_S$=-0.48). These results, while not conclusive, are broadly consistent with Jasechko *et al.* (2016)'s
observation of a significant negative correlation between $F_{yw}$ and average catchment slope in their survey of 254 global rivers. Jasechko et al. (2016) argued that this decrease in young water in steep terrain may be related to subsurface storage and transport. Deep vertical infiltration is prevalent in steeper landscapes; this is also consistent with conceptual models of groundwater flow (Gleeson and Manning, 2008) showing that larger topographic gradients lead to longer subsurface flow paths. Geotechnical stresses in steep terrain also promote fracture opening and thus increase hydraulic bedrock permeability
(Jasechko *et al.,* 2016; Gleeson *et al.*, 2011a), thereby facilitating deep percolation. On the other hand, catchments with steeper than 40° slopes are mainly found at higher elevations and thus tend to be snow-dominated. Here, smaller fractions of $F_{new}$ and $F_{yw}$ are expected due to snow storage.

We found no significant correlations between $F_{new}$ and either the fraction of area with potentially karstified carbonate
sedimentary rocks or the fraction of area with unconsolidated debris deposits. This result is somewhat surprising. One would expect that karstified rocks would increase $F_{new}$ by providing quick flow paths that allow bedrock storages to drain quickly (Hartmann et al., 2021). Furthermore, multiple previous studies have highlighted the importance of unconsolidated debris deposits in streamflow generation processes (e.g., Hood and Hayashi, 2015; Hayashi, 2019; Cochand *et al.*, 2019; Floriancic *et al.*, 2018). Our analysis is based on the global GLiM database, which uses lithological classes that may be too coarse, or
may be mapped at insufficient resolution, to be hydrologically informative at such rather small scales. Further analyses must await more comprehensive data on subsurface geology and near-surface geomorphological features (Gentile *et al.*, 2023; Floriancic *et al.*, 2022).

We also found a weak (but marginally significant) positive correlation between $F_{new}$ and the fraction of catchment area
covered by forests. von Freyberg *et al.* (2018) found similar correlations with $F_{yw}$ across Swiss headwater catchments, but Hrachowitz *et al.* (2021) found the opposite relationship for a forest removal experiment. It is somewhat surprising that the prevalence of forests is linked with higher values of $F_{new}$. A possible explanation might be tree roots facilitating preferential pathways through macropores (Brantley *et al.*, 2017; von Freyberg *et al.,* 2018), thus increasing $F_{new}$ and $F_{yw}$. However, these correlations might also be an artefact of cross-correlations with other variables, because the fraction of catchment area
covered by forests is also correlated with mean elevation and fraction of catchment area with slopes steeper than 40° ($R$ = -0.42 and -0.33, respectively).



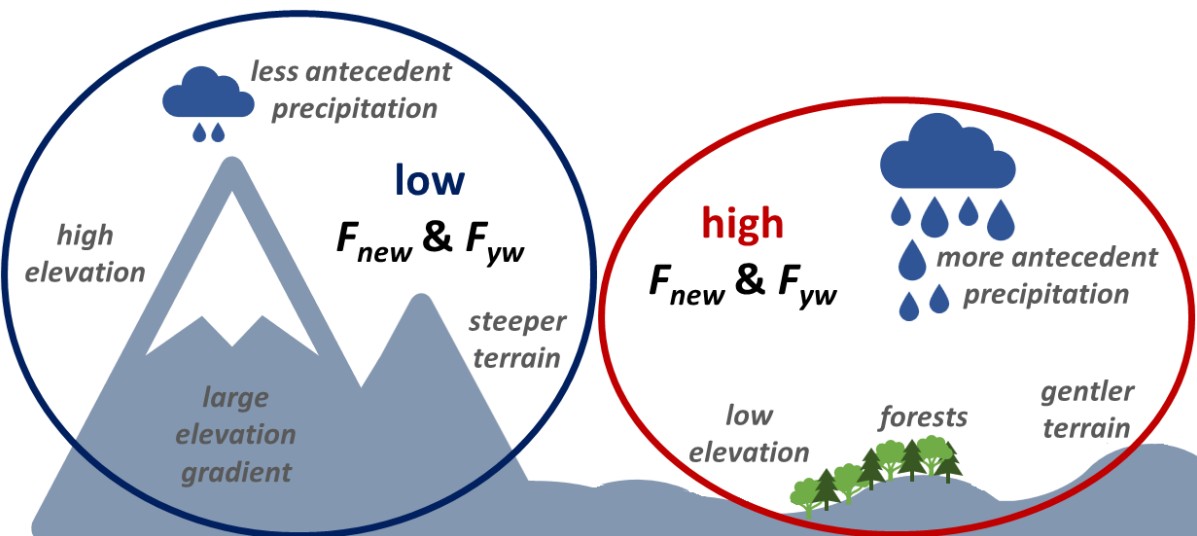

**Figure 10: Conceptual scheme of the significant relations of hydroclimatic variables and physical catchment properties to volume weighted new water fractions ($F_{new}$) across the 32 Alpine catchments.**

## 5. Summary and Conclusions

The Alps are an important water source for Europe, and assessing the partitioning of new (or young) versus old waters in Alpine rivers may provide important insights for sustainable management. For this study, isotope time series from 32 catchments across the Austrian and Swiss Alps were evaluated using two recently developed methods: estimation of new water fractions ($F_{new}$ – in this study, the fraction of water younger than about 1 month) using ensemble hydrograph separation, and estimation of young water fractions ($F_{yw}$ – the fraction of water younger than 2-3 months) using seasonal isotope cycles. Mean $F_{new}$ and $F_{yw}$ decreased with mean catchment elevation and varied across precipitation regimes (Figure 3). Overall, $F_{new}$ was higher in the summer months and higher following months with higher precipitation (Figure 4). However, strong increases in $F_{new}$ were only observed in 8 of our 32 catchments, and only above a precipitation threshold of roughly 175 mm month$^{-1}$. For most catchments, $F_{new}$ remained below 10% even when precipitation inputs were large (Figure 5). $F_{new}$ decreased from the headwater streams to the large downstream basins of the Danube and Rhine, potentially as a result of natural dampening by larger source areas and lakes, as well as anthropogenic influences from dams and reservoirs (Figure 6). High $F_{new}$ are more likely in small catchments, at low elevations, with small elevation gradients, in catchments with larger forest cover and when precipitation is high (Figure 4, Figure 8), whereas low $F_{new}$ are more likely in large catchments, at high elevations, with large total relief, high baseflow (Figure 7, Figure 8), and low antecedent precipitation (Figure 4). The obtained results reveal which Alpine areas transmit recent precipitation more rapidly to runoff.



The analysis also highlights the importance of further research on the effect of snow processes on partitioning of new (or young) and old waters, as well as the need for higher-resolution lithological information.


**Data availability:**

Daily discharge time series for 12 of the 20 Swiss sites were obtained from the CH-IRP database (Staudinger *et al.*, 2020). Discharge time series for the remaining 8 Swiss sites and for all 12 Austrian sites were obtained from the Federal Office of the Environment "Hydrological Data and forecasts" database (FOEN, 2022a) and the "Hydrographisches Jahrbuch"

contained in the WISA database (Umweltbundesamt, 2022a), respectively. Daily catchment averaged precipitation was obtained from the gridded precipitation dataset E-OBS (version 20.0e) at 0.1-degree resolution (Cornes *et al.*, 2018), *PET* was calculated from the "Global Aridity Index and Potential Evapotranspiration Climate Database v2" (Trabucco and Zomer, 2019);

Monthly gridded precipitation isotope reanalysis database Piso.AI (Nelson *et al.*, 2021) accessible at

https://isotope.bot.unibas.ch/PisoAI/ Monthly streamflow isotopes for the 12 Austrian sites were obtained from the WISA "H2O Fachdatenbank" database (Umweltbundesamt, 2022b) and for 8 Swiss stations from the NAQUA ISOT ("Nationalen Grundwasserbeobachtung – Isotopendaten") database (FOEN, 2022b). Streamflow isotopes for 12 additional stations across the Swiss Alps were obtained from the CH-IRP database (Staudinger *et al.*, 2020).

**Author contribution:**

MF designed the study, collected the data, ran the calculations, and wrote the first draft. All authors provided feedback during the analysis and preparation of the manuscript.

**Competing interests:** The authors declare that they have no conflict of interest.


**Acknowledgements:**

The authors would like to acknowledge the institutions and individuals that collected the long-term isotope data that made this study possible: WISA "Wasserinformationssystem Austria", ISOT - FOEN "Swiss Federal Office of the Environment" and the team of University of Zurich and University of Freiburg (Maria Staudinger and colleagues).

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
