# Peer review of "New water fractions and their relationships to climate and catchment properties across Alpine rivers"

_EGUsphere, 2023_

## Community Comment (CC1)

**Overview**

Floriancic et al. analysed how spatial and temporal patterns of water age across 32 Alpine rivers are related to hydroclimatic and physical catchment properties by calculating young and new water fractions from already existing isotope timeseries of $^{18}O$ and $^{2}H$. Lower young and new water fractions were observed in catchments with higher mean elevation, steeper slopes, larger catchment area, less antecedent precipitation as well as in catchments dominated by snow or affected by large water reservoirs such as lakes and dams. It is one of the first studies to use new water fractions on a regional scale, which has promising potential in improving the understanding of runoff generation processes. However, many of the observed relationships do not significantly advance on previous research and the study does not convincingly identify the dominant process controls, which even leads to some contradicting claims. Interesting results are found in exploring subsets of data, which is made possible by the novel methodology and would deserve a publication, although the paper overall requires a more detailed discussion and justification of certain analyses.

**Main comments**

1) Cross-correlations leading to spurious conclusions

Much of the paper is dedicated to correlating new water fractions to a range of hydroclimatic variables and physical catchment properties, which are highly cross-correlated and thus lead to issues in identifying the driving controls. This issue has already been long acknowledged and several similar relationships of young water fractions to catchment properties have been published on this topic (e.g. Jasechko et al., 2016 or von Freyberg et al., 2018). It is unclear from the text why new water fractions should have different relationships to catchment properties than young water fractions and how repeating such a broad analysis contributes to the process understanding. Recent research has instead focused on identifying individual factors within this variability such as the effect of snow cover (Gentille et al., 2023). The significant influence of the hydroclimatic regime has been confirmed even in this study, showing that new water fractions are much lower in snow-dominated catchments where snowpack provides longer water storage. Yet this effect is neglected when exploring the downstream propagation of new water fractions (which is presented as one of the focal points of this study) because sub-catchments of the Rhine and Danube are sorted by area without considering their location and climate. Due to sampling bias, many of the smaller catchments of the Rhine are rain-dominated, while snow-dominated catchments make up four of the six catchments with largest areas. This leads to a spurious and contradicting conclusion about the effect of catchment area because catchments typically shift from being snow-dominated to hybrid and rain-dominated along a single river. Mixing these effects could be avoided if only sampling locations along a single river (and not its tributaries) were considered.

The Danube case is slightly more convincing, although a statistically significant correlation is obtained only after the removal of two snow-dominated headwaters and with the acknowledgment that the three most downstream sampling locations are largely affected by dams. Thus, the main conclusion is a confirmation that dams indeed lower new water fractions without being able to say much about the effect of catchment area which could be transferrable to other rivers. I would suggest that the authors consider the perceptual model of downstream propagation of water age outlined by Gentille et al. (2023) based on the shifts in hydroclimatic regimes with low young water fractions expected in high-elevation snow-dominated headwaters, higher fractions in lower hybrid catchments and low fractions in downstream rainfall-dominated reaches. To observe such a relationship, a more sophisticated statistical method than a single Spearman correlation covering the whole dataset would be needed. Ideally, the relationship to catchment area would be derived from a completely rain-dominated catchment to eliminate the influence of hydroclimatic regimes altogether.

**2) Superficial analysis of data subsets**

I see the most significant contribution of this study in section 3.4 comparing different subsets of data, which takes advantage of ensemble hydrograph separation as opposed to other approaches and deserves a more detailed analysis. The study finds that new water fractions strongly increase after a precipitation threshold of 70 to 225 mm in most catchments. It would be interesting to see a discussion of what might be causing these differences in precipitation thresholds, possibly by relating it back to the catchment characteristics. It might also be worth focusing on how the precipitation threshold of 150-200 mm/month for most catchments relates to the 5 mm/day threshold observed by Knapp et al. (2019) because it appears as almost linear extrapolation of the daily precipitation intensity to the monthly scale at first glance, although the distribution of the precipitation during the month might influence the relationship. Additionally, ensemble hydrograph separation could be exploited further by distinguishing between winter and summer precipitation, which could yield considerably different thresholds in hybrid (and possibly snow) catchments according to Gentille et al. (2023). Such an effect is also suggested by observing higher new water fractions for the summer half of the year than for the wettest half of the year, indicating that precipitation in winter behaves differently (probably due to snowfall and snowpack storage in some catchments), which could be analysed further. Furthermore, monthly isotope timeseries would be appropriate to analyse more detailed patterns of seasonality or wetness conditions than just splitting the dataset into two halves (winter and summer, wet and dry) which averages over too much of the dataset to distinguish the control processes.

Concerning the interpretation of the catchments which do not experience increases in new water fractions with higher precipitation, it appears much of it can be explained by hydroclimatic regimes and catchment specificities instead of just stating relationship to elevation and slope. Unsurprisingly, all the catchments dampened by upstream lakes (INE, AAT and RHW) had low new water fractions no matter the precipitation magnitude. And the other catchments without precipitation thresholds are mostly snow-dominated catchments, which can be related to the effects of snow on water age as discussed by Gentille et al. (2023).

It should also be noted that the results of this section are confusingly interpreted in the discussion and conclusion where only 8 instead of 18 catchments are reported to have the precipitation threshold effect, overturning the whole message.

**3) Potential issues of data compatibility**

As the study relies on secondary data, compatibility of the individual datasets should be elaborated on more to strengthen the conclusions. Notably, the CH-IRP dataset was obtained at a fortnightly resolution (Staudinger et al., 2020), whereas WISA and ISOT provide streamflow isotope sampling data only at a monthly resolution. The methodology does not explain how this discrepancy was dealt with, even though higher sampling frequency leads to higher young water fractions (Stockinger et al., 2016) and lower new water fractions (Knapp et al., 2019) and thus potentially spurious comparisons. Information on the data sampling period is also missing (in Table 1) so it is unclear what years does the analysis cover and if it is the same for all locations (only number of samples per location is provided).

It is good to see use of a recently developed precipitation isotope reanalysis dataset which seems to have promising accuracy (Nelson et al., 2021), however, the study could be strengthened by providing a validation of the dataset against the sampling locations within the catchment boundaries, if possible, as this could give indication about the uncertainty arising from this factor. Furthermore, averaging the precipitation isotopes over the catchment area could lead to inaccuracies in large catchments with high precipitation gradients. It could be beneficial to weigh the averaging by mean precipitation to get a better representation of the relative contribution of precipitation to streamflow in different parts of the catchment or at least provide a sensitivity analysis of the weighting.

**Minor comments**

Abstract

The abstract covers slightly different results than the conclusion. Notably, no results about the precipitation thresholds are presented here despite being an important conclusion.

Line 21: Highest new water fractions (9.6%) were found in hybrid (not rainfall-dominated) catchments according to the results. Means across all catchments for young and water fractions were never presented in the text in results.

Line 26: Missing word – the fraction of slopes steeper *'than'* 40°.

Line 28: Replace elevation gradients by elevation difference or relief.

Introduction

The introduction provides a good background about previous studies of water age in rivers, although new water fractions (and their benefit over other measures) should also be introduced in this section already since they are part of the research questions. Hypothesis about the downstream propagation of new water fractions would also be welcome.

Line 107: The reference should be *(Kirchner and Knapp, 2020b).*

Line 109 (Research Question 1): The absolute value of new and young water fractions across Alpine rivers does not provide much information if it is known that the value sampled at a monthly resolution might be different from those sampled at higher frequencies, the observed values are also never related to other studies in the discussion. The question rather appears to be answered primarily by comparing young and new water fractions across different hydroclimatic regimes, hence it would make more sense to formulate the question according to that.

Line 113 (Research Question 3): Is monthly precipitation total really a measure of precipitation intensity? Consider rephrasing or going deeper in the analysis of precipitation distribution in the month (could potentially be based on daily streamflow or rain gauge records).

Methods and Available Data

Section 2.1 and 2.2: Missing information on the time period of obtained datasets (provided only for the precipitation dataset).

Figure 1: It would be more helpful to provide site codes in the map rather than stating their coordinates in Table 1.

Section 2.2 and 2.3: How were hydroclimatic regimes classified? Does it follow Weingartner and Aschwanden (1992) in line with similar publications? Motivation of studying relationships to all catchment properties (e.g. PET or elevation difference) should be mentioned in the methodology or introduction.

Lines 223-229 belong to Physical catchment properties. Missing justification of using fractions of slopes below 10° and above 40° (arbitrary value). Only six topographic properties are mentioned, although seven is written on line 224.

Line 236-238: Explanation of boxplots is redundant here.

Results

Table 2: The column $q\ P^{-1}$ is presented as fractions but with the units of %.

Line 258: Not consistent in use of ' in large numbers in different parts of the text and in figures.

Line 306: It should be specified more clearly that Pearson correlation was used for the relationship between new and young water fractions if that is the case or use $r_s$ to signify Spearman rank correlation.

Figure 3a, 3b and Figure 6d: Two catchments are labelled as AAB, although one of them corresponds to AAR instead.

Figure 4: Rainfall and hybrid catchments cannot be easily distinguished; different colour choice would make the figure clearer.

Line 334: Catchment DOW appears in both 175 and 200 mm threshold groups, should be replaced by ALP in the 200mm group.

Line 394: Excess word *'or'*.

Lines 411-416 repeat methodology and can be omitted. This section mentions ratio of elevation difference to catchment area which has not been used anywhere else. While it might be a better indicator than elevation difference itself (which is inherently related to catchment area), how does this measure differ from mean slope?

Figure 7 and 8: It would be interesting to see the effect of hydroclimatic regimes here the by plotting them in different colours such as in Figure 4.

Discussion

In general, results are well related to previous studies and expected physical drivers, however, the processes possibly causing the discrepancies of the results are mostly not explained in enough detail. It would also be beneficial to provide links to more studies using different measures of water age than young and new water fractions.

Line 446: How do you explain the opposite correlation compared to von Freyberg et al. (2018) despite using part of the same dataset? Do you get the same correlation for the 12 overlapping catchments as was found in their study?

Line 448: Rain-dominated and hybrid catchments appear to have similarly high values (hybrid catchments even have higher new water fractions than rain-dominated), only snow-dominated catchments have significantly lower values.

Line 452: Provide process explanation of why catchment size should matter in explaining the discrepancy with Gentille et al. (2023).

Line 469: Since correlations tend to improve when catchments with large lakes are removed (Jasechko et al., 2017), would removing these catchments have a significant impact on the derived correlations?

Line 505: What could be driving the discrepancy between the expected effect of evapotranspiration and the results? Can you imply that the effect of PET is not as significant as the other variables?

Line 509: Excess words *'were found'*.

Line 515-519: These results were not presented in the results section.

Lines 517 and 550: Correlation should be signed $r_s$ instead of R to be consistent.

Line 545: Von Freyberg (2018) should start with a capital V at the beginning of the sentence.

Figure 10: Use elevation difference or relief instead of elevation gradient.

References

Data sources are not properly referenced (e.g. Umweltbundesamt, 2022 or FOEN, 2022).

Line 695: Incomplete reference

---

## Community Comment (CC2)

**Review on New water fractions and their relationships to climate and catchment properties across Alpine rivers by Marius G. Floriancic, Michael P. Stockinger, James W. Kirchner, Christine Stumpp**

Reviewer: Arnaud Jansen
Student at Wageningen University

**Overview:**

The research in the paper is on the partitioning of new (younger than 1 month old) and young (2-3 months old) water in Alpine rivers in relation to hydroclimatic variables and physical catchment properties. The authors mention that the main importance of the research is to get a better understanding of the partitioning of old and new water so that it can lead to more sustainable water management. The research has been conducted over 32 Alpine catchments. The fraction of new water was determined using ensemble hydrograph separation and the fraction of young water using seasonal isotope cycles. After these fractions have been determined, the authors investigate whether there are explaining hydroclimatic variables or physical catchment properties that show a high correlation with the fraction of new and young water. The obtained results reveal which Alpine catchments transmit recent precipitation more rapidly to runoff. The paper concludes with a conceptualization of the relationship between young and new water fractions with hydroclimatic variables and physical catchment properties.

**Recommendation to the Editor:**

In the past, studies have mainly focussed on linking new water fraction to hydroclimatic variables and physical catchment properties for small headwater catchments (Gentile et al., 2023; von Freyberg et al., 2018; Ceperley et al., 2020). The main novelty of this research is that the authors have systematically linked new water fractions to hydroclimatic variables and physical catchment properties for both smaller headwater and larger downstream catchments. Furthermore, most previous studies on the partitioning of streamwater have only focussed on the fraction of young water. Together with few earlier studies (Knapp et al., 2019; Kirchner, 2019), this study is pioneering how new and young water fractions together can help improve the understanding of the partitioning of new and old water. These are not major novelties, but still relevant to have an impact on the field of catchment hydrology.

The paper is well written and statistics have been integrated well. The research is being portrayed in a broader perspective as they compare their findings to earlier research. The study addresses some important problems in the analysis of (isotope) data. For example, impact of snow, impact of elevation on precipitation isotopes and spatial resolution of the lithology map. However, there is limited discussion what has been done to validate whether these problems (and other assumptions made) create a significant bias in the results. Finally, there are certain parts that need clarification, especially on methodological and conclusions sections.

To conclude, I think this article could have an impact in the field of catchment hydrology, but it is not major. The paper fits well in the context of the journal. I believe it can be published with minor revisions.

**Major comments:**

**Major issue 1:** In the introduction the authors stated the following: "Although most multi-catchment time series have been sampled at low temporal frequency, they can nonetheless be used to assess the mixture of streamflow sources on time scales similar to their sampling intervals." This assumption justifies whether a lower temporal resolution of 1 month can be used for the purpose of this research. I believe this to be a bold statement. The authors back their statement with a paper by Hrachowitz et al. (2009). This paper, however, looked at data with weekly and fortnightly time series instead of a monthly time series. This is a significant difference. Moreover, they did not look into $\delta^{18}O$ and $\delta^2H$ isotopes but into $Cl^-$ concentrations as a tracer.

Some processes that determine the partitioning of young and old water in river discharge have an effect at significantly smaller time scales than at a monthly time scale. For example, earlier research suggested that there is significant correlation with distinct storms/precipitation events, meaning that these processes have an effect on a shorter time scale than one month (Knapp et al., 2019). Findings by Knapp et al. (2019) show strong increases in 7-hourly and weekly new water fraction above a precipitation threshold of roughly 5 mm day$^{-1}$. This entails that water partitioning might behave very differently for two different scenarios with the same rainfall intensity averaged over a month. For example, one single heavy rainstorm in a month or multiple light drizzles might lead to a different partitioning of new and old water. Moreover, the streamflow isotopes were measured at one moment in time in the month. This means that the authors assume that precipitation intensity at the beginning of the month contributes the same to monthly streamflow isotopes as more recent precipitation intensity days prior to the isotope streamflow measurement.

Thus, if processes with a time scale shorter than 1 month have a significant influence on new water fractions in streamflow (which research suggests), there can be a significant bias in the correlations computed for processes similar to or larger than the sampling interval. I believe that this should be made more explicit when making this statement. I would like the authors to go more into depth about the uncertainties, regarding the temporal scale, and the effect it might have on the results.

**Major issue 2:** Secondly, the authors mention that precipitation isotope data should not be interpolated between individual station measurements to determine precipitation isotopes for the different catchments. This is due to the fact that isotope fractions change with altitude and other factors. This is why they make use of the monthly gridded isotope model framework Piso.AI which makes use of machine learning (Nelson et al., 2021). Furthermore, they averaged the values of these grids within the boundaries of each of the catchments.

Even though this method is probably an improvement over interpolating between measurement stations, I believe it is likely still not accurate enough to precisely estimate precipitation isotopes for the purpose of this research. The resolution of 0.5° is quite course and precipitation isotopes can differ considerably spatially in mountainous areas. Jouzel et al. (2000) argue that higher spatial resolution of models helps significantly to increase the agreement of simulated precipitation isotope patterns with observations. Also, this study looks into some small catchments. Averaging precipitation isotopes on a spatial resolution that is bigger than the catchment size does not seem to give a realistic estimate to me. Moreover, the model framework has only been validated generically and not with a special focus on mountainous areas (Nelson et al., 2021).

I would recommend validating the Piso.AI data with the available data. As the precipitation isotope data gets averaged over the catchment area, I would suggest investigating the distribution of the

measuring location used in Piso.AI. If these locations happen to be located near the mean elevation of the larger grid, there is a good reason to believe the precipitation isotope data is representative for the average precipitation isotope fraction over the area. If the authors cannot validate the data, I would like the authors to stress the effect the uncertainty in precipitation isotopes has on the results. If one uses precipitation isotope data with a significant error, fractions of new and young water in streamflow are calculated with a bias. This will show in the eventual correlations between new and young water fraction and hydroclimatic variables and catchment properties.

**Major issue 3:** Furthermore, I think some of the claims made in the conclusion are not supported well enough with evidence. For instance, I do not believe that the correlation between the fraction of young water can be described by the amount of forested area in the catchment. I believe that this is probably an example of cross-correlation. Forest might just happen to thrive better at certain areas that favour quick run-off (e.g., certain slope, elevation). At higher altitudes there is limited to no forest cover in the Alps. As the new water fraction is lower at higher altitudes (probably due to other factors), it is likely to find a positive relationship with forest cover. *Figure 9* supports my suspicion as there seems to be a significant correlation with elevation. The authors do shortly touch upon this in the discussion, but seem to discard this in the conclusion. Moreover, the authors mention that Hrachowitz et al. (2021) found the opposite relationship for a forest removal experiment. The authors give no explanation why these two different studies could contradict each other. If there seems to be contradicting results between studies, I suggest being more hesitant to draw conclusions. I would recommend removing this claim from the conclusions.

**Minor comments:**

**Minor issue 1:** I believe the title of the paper can be polished a bit. The title at the moment is: "New water fractions and their relationships to climate and catchment properties across Alpine rivers." The title could also be interpreted in a different way. One could also think this paper is about changing discharge patterns due to changing climate and catchment properties. I can understand why the authors chose the word 'new', as it is one of the first studies to look into the water fraction younger than one month old, as previous research looked into young water that was younger than 2-3 months. Also, previous studies (Knapp et al., 2019) used a similar title. My suggestion would be to change the title to "New and young water fractions…", as the paper looks into both new and young water fractions.

**Minor issue 2:** The authors mention the knowledge gap, but it would be clearer if they would also explicitly mention what the aim of the research is. The aim could be along the following lines: *To get a better understanding of how much streamflow is derived from old water stored in the subsurface, versus more recent precipitation that reaches the stream via near-surface quick flow processes and also how this partitioning varies across different Alpine catchments in response to hydroclimatic forcing and catchment characteristics.*

**Minor issue 3:** Ensemble hydrograph separation has been used to calculate new water fractions in streamflow. However, the authors do not discuss potential biases in the result that this method might produce. I would suggest the authors to discuss in the methods section.

**Minor issue 4:** The authors try to correlate catchment slope to the fraction of new water in streamflow, as earlier research by Jasechko et al. (2016) suggested that there seems to be a strong negative correlation. The authors did not seem to find such a strong correlation between slope and new water fractions in streamflow in this research. I believe this is why they decided to look at slopes below 10° and above 40°. The reasoning why the authors picked these specific slopes is not mentioned. I understand that the authors look at areas with milder and very steep slopes, but the

number seems a bit arbitrary to me. Maybe the authors can explain how they got to these specific slopes.

**Minor issue 5:** The conclusions made in the conclusion do not match the conclusions made in the abstract. In the abstract they mention the differences in new water fractions between rainfall-dominated and snowfall dominated catchments. However, there is no mention of this in the conclusion. Also, the authors mention in the conclusion that that new water fractions decrease from headwater streams to large downstream basins of the Danube and Rhine, but this is not explicitly mentioned in the abstract. There are more examples of these which I will not mention. I would strongly suggest to align the conclusion stated in the abstract and conclusion.

**Minor issue 6:** The authors switch between 'new water fraction' and '$F_{new}$' through the course of the paper. I would settle on using one of the two. This also applies to the fraction of young water.

**Minor issue 7:** Most figures do not have a title/heading. I would suggest putting up headings for most figures (and sub-figures) as it makes it easier to interpretate the figures.

**Specific comments:**

P1-2, lines 29-32: this sentence is very long. It might be easier to comprehend when it is split up in two different sentences.

P3, lines 73-74: "… found although…'' this part of the sentence does not read well. Either put a comma after "found" or change the order of the sentence by rewriting it to: "In global-scale syntheses, Jasechko *et al.* (2016, 2017) found that 25% of global streamflow is younger than 1.5 - 3 months, despite most groundwaters are dominated by fossil waters" I would suggest the latter option.

P4, lines 109-118: I think the list of research questions could be better formatted. I do not see the need for the blank line in between the research questions.

P5, figure 1: please make a larger northern arrow. It would look better if the northern arrow scale bar and legend are properly aligned. Also, the city names are difficult to read. Please enlarge these.

P16, figure 5: the graph is messy which makes it difficult to read. The graph might become easier to read if the y-axis is set to a logarithmic scale.

P18, figure 6a: please add a legend to this map. I believe the legend might be the same as in figure 6b, but that is not clear. One could also combine the two figures in such a way that this is clearer. Also, the city names are difficult to read. Please enlarge these.

P18, figure 6b: please align the scale bar and northern arrow.

**References**

Ceperley, N., Zuecco, G., Beria, H., Carturan, L., Michelon, A., Penna, D., Larsen, J., & Schaefli, B. (2020). Seasonal snow cover decreases young water fractions in high Alpine catchments. Hydrological Processes, 34(25), 4794–4813. https://doi.org/10.1002/hyp.13937

Jasechko, S., Kirchner, J. W., Welker, J. M., & McDonnell, J. J. (2016). Substantial proportion of global streamflow less than three months old. Nature Geoscience, 9(2), 126–129. https://doi.org/10.1038/ngeo2636

Jouzel, J., Hoffmann, G., Koster, R. D., & Masson, V. (2000). Water isotopes in precipitation: data/model comparison for present-day and past climates. Quaternary Science Reviews, 19, 363–379.

Gentile, A., Canone, D., Ceperley, N., Gisolo, D., Previati, M., Zuecco, G., Schaefli, B., & Provenzale, A. (2023). Towards a conceptualization of the hydrological processes behind changes of young water fraction with elevation: a focus on mountainous alpine catchments. Hydrology and Earth System Sciences, 27(12), 2301–2323. https://doi.org/10.5194/hess-27-2301-2023

Hrachowitz, M., Soulsby, C., Tetzlaff, D., Dawson, J. J. C., & Malcolm, I. A. (2009). Regionalization of transit time estimates in montane catchments by integrating landscape controls. Water Resources Research, 45(5). https://doi.org/10.1029/2008wr007496

Hrachowitz, M., Stockinger, M., Coenders-Gerrits, M., Van Der Ent, R. J., Bogena, H., Lücke, A., & Stumpp, C. (2021). Reduction of vegetation-accessible water storage capacity after deforestation affects catchment travel time distributions and increases young water fractions in a headwater catchment. Hydrology and Earth System Sciences, 25(9), 4887–4915. https://doi.org/10.5194/hess-25-4887-2021

Kirchner, J. W. (2019). Quantifying new water fractions and transit time distributions using ensemble hydrograph separation: theory and benchmark tests. Hydrology and Earth System Sciences, 23(1), 303–349. https://doi.org/10.5194/hess-23-303-2019

Knapp, J. L. A., Neal, C., Schlumpf, A., Neal, M., & Kirchner, J. W. (2019). New water fractions and transit time distributions at Plynlimon, Wales, estimated from stable water isotopes in precipitation and streamflow. Hydrology and Earth System Sciences, 23(10), 4367–4388. https://doi.org/10.5194/hess-23-4367-2019

Nelson, D. B., Basler, D., & Kahmen, A. (2021). Precipitation isotope time series predictions from machine learning applied in Europe. Proceedings of the National Academy of Sciences of the United States of America, 118(26). https://doi.org/10.1073/pnas.2024107118

Von Freyberg, J., Allen, S. T., Seeger, S., Weiler, M., & Kirchner, J. W. (2018). Sensitivity of young water fractions to hydro-climatic forcing and landscape properties across 22 Swiss catchments. Hydrology and Earth System Sciences, 22(7), 3841–3861. https://doi.org/10.5194/hess-22-3841-2018

---

## Community Comment (CC3)

**Peer review "New water fractions and their relationships to climate and catchment properties across Alpine rivers" by Floriancic et al., 2023.**

The paper by Floriancic et al. (2023) aims to quantify the new and young water fractions in Alpine rivers and to link them to hydroclimatic drivers and physical catchment properties. Stable water isotope time series in precipitation and streamflow are used to determine the relative proportions of young (younger than 2-3 months) and new (younger than one month) water. Young water fractions ($F_{young}$) are determined based on seasonal precipitation isotope cycles, as summer precipitation is isotopically heavier than winter precipitation. New water fractions ($F_{new}$) have been determined based on hydrograph separation.

Overall, the paper is written well and the conclusion provides a clear overview of the research by answering the research questions as stated in the introduction. Besides, the introduction clearly explains the importance of knowing young and new water fractions in Alpine rivers. However, the novelty of the paper, other than using a range of catchment sizes, remains unclear to me, considering that there are multiple papers on the same topic (Knapp et al., 2019; Kirchner, 2019; Ceperly et al. 2020) some of which with partly overlapping study areas (von Freyberg et al., 2018; Gentile et al., 2023). Therefore, I would like to encourage the authors to expand the introduction by highlighting the novelty and scientific contribution of their research. The research fits the scope of the journal and is in my opinion suitable for publication after addressing the issues mentioned below.

**Major comments**

*Covariances influencing correlations*

In section 2.6 it is explained how the authors will draw conclusions based on statistical measures such as Spearman rank correlations and Wilcoxon Signed-Rank tests. One of the findings is a positive relationship between Potential Evapotranspiration (PET) and $F_{new}$, which is described as counterintuitive in the discussion section (Line 505) as one would expect the fraction of new water to decrease with increasing evapotranspiration. However, the authors do not go into this any further nor do they refer to other researches with similar or contradicting results. In section 4.2 Figure 9 shows the Spearman rank correlations between the selected hydroclimatic variables and physical catchment characteristics across the study area. In this figure a positive relationship can be seen between the fraction forest and PET. The authors found a (weak) positive correlation between $F_{new}$ and forest cover, which was also found by Freyberg et al. (2018). In the paper the authors suggest that forest might have a role in shaping flow paths and therefore influence water ages. This makes me wonder whether PET is actually influencing the amount of $F_{new}$ in the streamflow or whether it is the other way around; due to more forest $F_{new}$ is higher, but the forest will also lead to an increase in PET.

This leads to issue (1): the authors seem to overlook the fact that many things co-vary, other than stating, based on Figure 9, that it remains unclear which of the variables is a first-order control on new water fractions. Section 2.6 does not give a proposed mechanism to find out which variables have the largest impact on new water fractions. Without researching covariances, the question remains how useful this type of conclusions is, and to what extent it fulfills the aim of the paper to understand the origins of streamflow and to determine the influence of hydroclimatic variables and physical catchment properties. This issue could be addressed by extending the discussion section by reflecting

more on the implications of the results. As an alternative, the authors could extend the methodology by proposing a mechanism to investigate the control of variables on new water fractions ruling out covariances between the variables. Besides, correlations are more convincing when a plausible mechanistic explanation can be identified. Identifying the first-order control on new water fractions leads to more robust conclusions when mechanistic explanations can be provided as evidence for the found correlations.

*Robustness of results*
New water fractions are calculated using ensemble hydrograph separation, clearly explaining how this method is insensitive to unknown or unmeasured endmembers. Next, the authors explain the use of volume weighted new water fractions, which reduces the effect of catchment wetness. This is in agreement with von Freyberg et al. (2018), who use the same approach. Also, the dataset that is used is split up based on the monthly precipitation values to assess the differences in $F_{new}$ for the wettest and driest half of the year. This type of analysis will provide insight into the effect of climate and antecedent conditions on the fraction of new water. The issue (2) that rises is that, surprisingly, the authors did not take into account the effects of hydrograph shape, which is determined by catchment characteristics such as shape and slope, on the calculation of new water fractions. This might indirectly influence the reliable of the results through the calculations of $F_{new}$ across catchments and the therewith calculated correlations. Therefore, the robustness of the results drawn based on hydrograph separation is questionable.

Expanding the analysis by reducing the effect of hydrograph shape would improve the robustness of the results. Otherwise, the studied catchments could be selected such that they consist of similar catchment characteristics, but one characteristic. The influence of that certain characteristic on new water fractions can be researched, also addressing the issue of covariances. This however, would limit the research, as it aims to study the water fractions to hydroclimatic variables and catchment properties across small to very large basins. Besides, the correlations that are found might still not be causal, as variables could be overlooked.

*Spatial and temporal resolution of precipitation isotope data*
In section 2 the data collection is described. Precipitation isotope measurements of most catchments are available only at single sampling locations and the authors chose not to interpolate individual station measurements. Instead, issue (3), the precipitation isotope data is based on monthly gridded precipitation isotope reanalysis database by Piso.AI, averaged within the boundaries of the study catchments. The authors lack to mention the spatial resolution of the gridded precipitation isotope data, which undermines the decision to use the reanalysis database. Besides, the correctness of the precipitation isotope reanalysis should be evaluated, before a substantiated decision can be made to use the Piso.AI database.

Next, the use of monthly data, which is equal to the time scale over which water is new, might be too coarse to be able to say something about the fluctuations in $F_{new}$ following precipitation events. Knapp et al. (2019), who used 7 hourly and weekly timescales, observed stronger increases in $F_{new}$ as a result of precipitation, which leads to the question whether using monthly precipitation isotope data gives reliable $F_{new}$ values. The paper needs more in depth discussion on the consequences of using monthly precipitation isotope data on the outcome on the research, clarifying the choice to use this temporal resolution.

**Minor comments**

The fourth research question (Line 115 & 116) "*How do new water fractions propagate downstream from headwater catchments to the large basins of the Danube and Rhine catchments?*" This research question is not introduced in the introduction nor is it mentioned in the abstract. Subchapter "*Downstream propagation of $F_{new}$ in Danube and Rhine*" is not related to any other research chapters in the paper. Therefore, the question is secluded from the rest of the research. Please add the results from this chapter to the abstract and make sure to introduce the research question and mention its use in this paper. Otherwise, it might be better to remove the question and its results from the manuscript.

The title of the paper falls short on its content as it only new water fractions are mentioned. The calculation of young water fractions cannot be inferred from the title, even though it is part of the research. Consider changing the title to *"Quantifying young and new water fractions and their relationship to climate and catchment properties across Alpine rivers"*.

The conclusion ends with the statement that the analysis highlights the importance of further research on the effect of snow processes on partitioning of new (or young) and old waters. However, the authors do not elaborate on how this would improve the research and what they would expect to find. I would like to encourage the authors to go into depth on this a bit more, especially since Gentile et al. (2023) discusses the impact of snow on the seasonal cycle of young water fractions and Ceperley et al. (2020) provides insight into the effect of snow cover on young water fractions. How would the further research that is needed according to the paper complement this research?

**Technical comments**

Line 50: Reference IAEA, 2019a missing in reference list.

Line 51: Reference IAEA, 2019b missing in reference list.

Line 61: Reference Kirchner et al., 2023 missing in reference list.

Reference list not in correct alphabetical order.

**References**

Ceperley, N., Zuecco, G., Beria, H., Carturan, L., Michelon, A., Penna, D., Larsen, J., and Schaefli, B.: Seasonal snow cover decreases young water fractions in high Alpine catchments, Hydrological Processes, 34, 4794–4813,https://doi.org/10.1002/hyp.13937, 2020.

Gentile, A., Canone, D., Ceperley, N., Gisolo, D., Previati, M., Zuecco, G., Schaefli, B., and Ferraris, S.: Towards a conceptualization of the hydrological processes behind changes of young water fraction with elevation: a focus on mountainous alpine catchments, Hydrology and Earth System Sciences, 27, 2301–2323, https://doi.org/10.5194/hess-27-2301-2023, 2023.

von Freyberg, J., Allen, S. T., Seeger, S., Weiler, M., and Kirchner, J. W.: Sensitivity of young water fractions to hydroclimatic forcing and landscape properties across 22 Swiss catchments, Hydrology and Earth System Sciences, 22, 3841–640 3861, https://doi.org/10.5194/hess-22-3841-2018, 2018.

Kirchner, J. W.: Quantifying new water fractions and transit time distributions using ensemble hydrograph separation: theory and benchmark tests, Hydrology and Earth System Sciences, 23, 303–349, https://doi.org/10.5194/hess-23-303-2019, 2019.

Knapp, J. L. A., Neal, C., Schlumpf, A., Neal, M., and Kirchner, J. W.: New water fractions and transit time distributions at Plynlimon, Wales, estimated from stable water isotopes in precipitation and streamflow, Hydrology and Earth System Sciences, 23, 4367–4388, https://doi.org/10.5194/hess-23-4367-2019, 2019.

---

## Author Response (AR1)

**Dear Editor,**

**Here we provide a revised version of our manuscript. The revised paper contains the necessary changes to address the reviewers' and editors' comments on our previous submission. Below we provide a detailed point-by-point response (in bold) to those comments (*in italics*).**

**Sincerely,**

**Marius Floriancic, Michael Stockinger, James W. Kirchner & Christine Stumpp**
* * *
Reply to RC 1:

*Comments on 'New water fractions and their relationships to climate and catchment properties across Alpine rivers' by Floriancic et al., 2023.*

*General comments:*

*Understanding runoff generation processes in high mountain catchments is important as it is water tower for providing water for drinking, agriculture, and hydropower production. The main findings in this manuscript reveal that Alpine rivers tend to have larger new water fractions at low elevations, in flatter terrain and while in smaller catchments with large forest cover. The findings are interesting to scientific community, however, is also controversial in different perspective. As claimed by the authors, there seems less studies linking new water fractions to hydroclimatic drivers and physical catchment properties across small to very large basins. We know that the application of chemistry tracing methods is usually underlain by many basic assumptions, for instance, applicants must account for the heterogeneity in rainfall isotope referring to Pinder and Jones (1969). Hence, the methods adopted as well as the conclusions in this manuscript should be carefully discussed to justify the rationality of the methods and the reliability of this conclusions.*

**Thank you. In the manuscript we use two methods to assess the fraction of more recent precipitation in streamflow (calculation of "young water fractions" and "new water fractions"). For the assessment of young water fractions, we fit a sinusoidal curve to the typical long-term seasonal precipitation cycle; for the assessment of new water fractions we use the recently developed ensemble hydrograph separation method. Our results reveal the typical long-term averages of young and new waters in stream and do not reveal any information on short term variations as introduced by heterogeneities outside the bounds of the typical seasonal precipitation cycle. Unfortunately, measurements on isotope ratios in precipitation are rarely**

**available at high spatial resolution. Therefore, we used the available gridded precipitation dataset provided by Nelson et al (2021), where factors like altitude effects are considered. In the revised version, we added a discussion section on "Limitations in data availability" where we describe the limitations in more detail and compare different data products of the available precipitation isotope data (shown now in the supplement – Figures S5 & S6).**

**Importantly, both young water fractions and new water fractions do not depend on the absolute values of either precipitation or streamflow isotopes, but only on their variations over time (their seasonal cycles for young water fractions,  and their month-to-month fluctuations for new water fractions).  Thus any offsets introduced by terrain effects will have little or no impact on calculated young water fractions and new water fractions.**

Moreover, as there are cross-correlations between many catchment properties, and hydroclimates potential drivers, the explanations that which may be a first-order control on new water fractions should be further verified with more cautions as indicated by the authors. For example, the authors found that high fractions of new water (Fnew) were more likely in small catchments, at low elevations, with small total relief and larger forest cover, and following months with high precipitation. However, they also found that Fnew tended to decrease downstream, from smaller headwaters to larger river basins, in which it can be inferred that altitude, relief, slope gradient, and even precipitation (see in Ménégoz et al., 2020) all decrease with altitude from smaller headwaters to larger river basins in Alps. A reasonable explanation to the issue is that it is the impact of water storage in lakes and reservoirs, as well as the potential effects of anthropogenic flow regulation when moving from the headwaters downstream. But it seems a little bit contradiction in the context. So, the authors should provide more deepen and persuasive discussions for this key conclusion in the manuscript. I wonder whether water stored as snow or glaciers in high altitude basins have led to relatively lower Fnew in headwaters, which is quite inconsistent with our intuition about runoff generations. Cause in alpine basins more areas with naked rocks or thin soils could be observed in high altitude regions as erosion rates increase with local relief (Heimsath et al., 2012), thus, more rainfall directly drains to drainage networks as new fractions in high altitude areas?

**Thank you. In the revised version of the manuscript, we now provide a more detailed discussion of the dependencies between different catchment descriptors. Of course, we acknowledge that many catchment characteristics are correlated; this was the reason why we included a cross-correlation matrix in the manuscript (see Figure 9). Regarding the comment about the importance**

**of snow in higher-elevation catchments of the Alps we'd like to point out that this issue is not yet fully resolved; we reported this in the original version of the manuscript and now extended the discussion on this issue. It has not been fully understood yet to what extent the lower fractions of more recent precipitation found in Alpine streams is due to snow processes or due to the large elevation gradient (and thus large subsurface storage volumes). With our sample of catchments (as in many other studies before) we cannot test these hypotheses independently (catchments with large topographic gradient are also dominated by snow across the European Alps). Still, we adopted the discussion to point out that the smaller fractions of more recent precipitation in the higher Alpine catchments have multiple potential explanations but need to be considered in another study based on a targeted dataset.**

In addition, $F_{yw}$ as defined by the authors is the ratio of seasonal amplitudes of sinusoidal fits to precipitation and streamflow isotope time series. However, as shown in Figure 2, the seasonal fluctuations in streamflow isotopes among all the basins are quite similar. Therefore, Is $F_{yw}$ only related to the corresponding fluctuations in precipitation isotopes? We have known that interannual variations of precipitation isotopes have a significant impact on the young water fraction (Gou et al., 2023; Dai et al., 2022), which raises doubts about the results. Could it be consistent in the results obtained by using isotopic data with different time series lengths? Furthermore, the sampling frequency of precipitation isotopes also has a significant impact on the young water fraction (Gallart et al., 2020; Stockinger et al., 2016). So, is it reliable to estimate $F_{yw}$ through using monthly-scale data?

**The young water fractions can only be calculated as the amplitude ration between multiple years of precipitation data AND streamflow data; splitting of these data on an interannual basis should never be done as it will lead to flawed results (see the descriptions in Kirchner 2016). This was confirmed by the case studies of Gallart et al. (2020) and Stockinger and Stumpp (2024), as their studies found high uncertainty associated with $F_{yw}$ derived from one-year precipitation and runoff data, and Stockinger and Stumpp (2024) suggest a minimum time series length of six years for a robust, long-term $F_{yw}$ result. Also, it is important to point out that the sampling frequency does not impact the results, as long as the samples are bulk precipitation samples from the entire month and the isotopic signals are volume-weighed by the precipitation amount (which they are in the case of our study). The Stockinger et al. 2016 and Gallart et al. 2020 results do not indicate sensitivity on the sampling frequency of precipitation but rather of streamflow, and in any case**

**their results are primarily an artifact of the under-representation of high streamflow in the low-frequency sampling procedures that they used. Furthermore, since the precipitation and streamflow timeseries of all catchments are in monthly intervals, streamflow is biased towards lower streamflow values in all catchments. Thus Fnew at high flows can not be reliably estimated from monthly data, however we can infer the relationships between Fnew and hydroclimatic and physical catchment descriptors. It is important to point out that several published studies examine Fyw but do not use the method outlined by Kirchner 2016 as it was intended (and benchmark tested); thus, conclusions drawn from these studies should not be interpreted as reflecting on the original Fyw method.**

In general, the manuscript needs more deepen discussions, and the authors should provide more persuasive evidences to clarify the first order control of runoff generation across headwaters to down streams for credible conclusions.

**Thank you. We present all dependencies of hydroclimatic variables and physical catchment properties in Figure 9, and we also adapted parts of the discussion to stress the importance of the potential interdependencies of certain catchment characteristics i.e., we report the potential dependencies in the respective parts of the discussion and added a notice of caution for interpretation in the conclusions.**

Specific comments: (L means lines)

L121-131: added how precipitation varies with altitude.

**We present the correlation of mean, max and min elevation and annual, summer and winter precipitation in Figure 9, whereas correlations between elevation and precipitation are weak overall.**

L140-150: the accuracy of the monthly gridded precipitation isotope reanalysis database Piso.AI should be evaluated before adopted in the study regions.

**While we agree that the isotope reanalysis product Piso.AI is not the perfect, unfortunately it is the only gridded dataset that can be used for such large catchments. The use of point measurements is not reliable for such large basins, and other reanalysis methods (i.e., Seeger & Weiler 2016) yield similar results to Piso.AI. The gridded data and point measurements within our study area have been evaluated in great detail by Nelson et al. 2021. In addition, in the supplement of the revised manuscript we now show a comparison of data derived from the approach of Seeger & Weiler 2016 and the data from Nelson et al. 2021 for selected catchments (Figures S5 & S6), confirming that both approaches yield similar results.**

L258: The catchment areas range from 29 km2 to 103'946 km2. Such huge difference in area would lead to quite different patterns of rainfall and discharge distribution in different catchments which dramatically impact isotope fractions in downstream rivers, and weaken the basic assumption for isotope estimations, and there is especially large spatial heterogeneity of rainfall and discharge concentration across a catchment with larger areas. So how do you calculate monthly precipitation isotope particularly in catchments with larger area (e.g., >1000 km2)?

**As outlined in the manuscript, we use the gridded dataset of Nelson et al. 2021 that is to date the most reliable available product that allows the estimation of precipitation isotope ratios for larger basins. We emphasize this now in the revised manuscript and discuss potential uncertainties in the additional discussion section 4.1.**

*Technical corrections:*

*(1) In figure 1 Gauge names should be provided and drainage areas should also be added and listed in table 1.*

**We added gauge names in Figure 1 of the revised manuscript. Drainage areas are already available in Table 3.**

*References:*

*Dai, J., Zhang, X., Wang, L., Luo, Z., Wang, R., Liu, Z., ... & Guan, H. (2022). Seasonal isotopic cycles used to identify transit times and the young water fraction within the critical zone in a subtropical catchment in China. Journal of Hydrology, 612, 128138.*

*Gallart, F., Valiente, M., Llorens, P., Cayuela, C., Sprenger, M., & Latron, J. (2020). Investigating young water fractions in a small Mediterranean mountain catchment: both precipitation forcing and sampling frequency matter. Hydrological Processes, 34(17), 3618-3634.*

*Gou, J., Qu, S., Guan, H., Shi, P., Zhang, Z., Yang, H., ... & Han, X. (2023). Seasonal variation of transit time distribution and associated hydrological processes in a Moso bamboo watershed under the East Asian monsoon climate. Journal of Hydrology, 617, 128912.*

*Heimsath, A., DiBiase, R. & Whipple, K. Soil production limits and the transition to bedrock-dominated landscapes. Nature Geosci 5, 210–214 (2012). https://doi.org/10.1038/ngeo1380.*

*Ménégoz, M., Valla, E., Jourdain, N. C., Blanchet, J., Beaumet, J., Wilhelm, B., Gallée, H., Fettweis, X., Morin, S., and Anquetin, S.: Contrasting seasonal changes in total and intense precipitation in the European Alps from 1903 to 2010, Hydrol. Earth Syst. Sci., 24, 5355–5377, https://doi.org/10.5194/hess-24-5355-2020, 2020.*

*Pinder, G., Jones, J., 1969. Determination of the ground-water component of peak discharge from the chemistry of total runoff. Water Resour. Res. 5 (2), 438–445.*

*Stockinger, M. P., Bogena, H. R., Lücke, A., Diekkrüger, B., Cornelissen, T., & Vereecken, H. (2016). Tracer sampling frequency influences estimates of young water fraction and streamwater transit time distribution. Journal of hydrology, 541, 952-964.*

**Reply to RC 2:**

*The work of Floriancic et al. (2023) addresses the estimation of new water fractions with the ensemble hydrograph separation in 32 catchments to investigate their relationships to climate and catchment properties across Alpine rivers. The work is novel since, according to my knowledge, there are no previous studies that have related "new" water fractions with hydroclimatic and physical*

*properties. Nevertheless, there are some major issues that should be taken into consideration before considering the manuscript for publication in HESS.*

**Thank you for the positive feedback.**
* * *
*Main comments:*

*1) You have used monthly streamflow isotope for 12 Austrian sites and 8 Swiss stations. Moreover, you have used monthly gridded precipitation isotopes from Nelson et al. 2021. It is well known that the sampling resolution affect the young water fraction estimates (Gallart et al., 2020; Stockinger et al., 2016). When you compare your results with those of previous studies you should include the effect of sampling resolution since some discrepancies between your results and previous results can be also due to the low sampling resolution of isotope data used in your study. Additionally, by considering this sampling resolution you consider as "new" some water that is younger than 1 month. This threshold age is of the same magnitude of the typical threshold age for young water (2-3 months), as also reflected by similar Fyw and Fnew for some catchments very close to the 1:1 line in Figure 3. Accordingly, I would underline this when you state that similar relationship to catchment properties is obtained with young water fractions. This happens since they have a similar threshold age. I would add in the conclusion that in future studies is necessary to investigate the relationship between new water fraction and climate/catchment properties by using high-resolution (e.g., daily) isotope data (and accordingly much lower threshold ages). Please, specify in the manuscript title the age of new water: as soon as I read the title I thought that you have investigated event water with a threshold age of the order of few days.*

**Thank you. We changed the title to "Monthly new water fractions and their relationships to climate and catchment properties across Alpine rivers" to avoid this misconception and added that the ensemble hydrograph separation method (calculation of $F_{new}$) in the presented dataset allowed us to calculate the amount of streamflow that is younger than one month. Thus, in the manuscript we consider all water that is "new" to be younger than one month. However, it is important to mention that the sampling frequency of precipitation isotopes does not impact the _young_ water fraction ($F_{yw}$) results, as long as the samples are bulk precipitation samples from the entire month and the isotopic signals are volume-weighed by the precipitation amount (which they are in the case of our study). In the Gallert and Stockinger studies, the main difference between the high-frequency and low-frequency sampling was that the low-frequency sampling**

**under-represented high-streamflow conditions. Thus, given the well-known discharge sensitivity of Fyw (which itself is a key characteristic of catchment transport behavior), when high-streamflow conditions were sampled more often, Fyw understandably increased. Thus, the issue is not a bias in the Fyw method, but instead under-representation of high-streamflow conditions in the low-frequency sampling used by Gallert and Stockinger.**

2) About your conceptual scheme reported in Figure 10. I think that it is biased from the use of very large catchments. Indeed, some catchments you have used in your dataset, due to their large extension, cover the entire landscape (reported in Fig. 10) from high to low elevations (in some catchments the elevation difference is higher than 4000 m) including both steeper and gentler terrain, largely heterogenous land cover etc. Thus, the fraction of new water you estimate is the result of hydrological processes occurring across the entire catchment area. Indeed, Fnew could depend more on processes occurring at higher elevations or more on processes occurring in plain areas (e.g., with influence of natural and/or artificial lakes). In this regard I would suggest looking at your results by not considering very large catchments and to better detail your conceptual scheme by subdividing it in two/three different elevation ranges (e.g., a possible elevation threshold could be 1500 m a.s.l. since previous studies (e.g., Ceperley et al. 2020) have found a change in hydrological regime above this threshold).

**Thank you. It was one main aim of this study to compare catchments of different scales to understand to which extent fractions of more recent precipitation are dominated by catchment characteristics in a wide range of different-sized catchments. Certainly, larger catchments integrate over many processes, and we do not normally know which features predominantly control Fyw and Fnew. Still, our specific analysis on how Fnew changes along large streams indicates the dampening effects with increasing catchment size. While differences between catchments exist (also in studies that only consider small catchments) we think it is interesting to see that most of the established relationships between young & new water and catchment descriptors hold both for small and for large catchments. Therefore, the figure appropriately combines our findings across smaller and larger catchments.**

3) The database for geology is very coarse, but I guess nothing is available. Is it possible to better discuss its implications?

**Unfortunately, this is true for many studies, including ours. More systematic databases for geological information would be a big help, and we support any efforts in this direction. Even the CAMELS-CH dataset, only covering Switzerland, does not have better-resolved (and homogenized) geology information.**

4) In the paper you cited in relation to quaternary deposits, Gentile et al. (2023) in this same journal, it has been discussed the relation between baseflow and Fyw. Can you compare their results with yours, although the different size of the catchments?

**Thank you for this remark. We show the relation of $F_{yw}$ and $q_{95}$ in the supplementary material and found similar negative correlations as Gentile et al. 2023. We added the refence to Gentile et al (2023) to the discussion.**

5) Less important (the paper structure is a personal choice) : you have chosen to separate results from discussion. Nevertheless, I think that in many parts of the "Results" section you have also discussed the results. Moreover, in the "Discussion" section there are many parts that are redundant since you recall the results or methods sections. Accordingly, I would suggest changing the paper structure in "Results and Discussion". This will shorten the manuscript length and improve its readability.

**Thank you, we discussed this internally prior to the first submission and decided to go for a separate results and discussion section for the manuscript. We now reconsidered this choice but decided to stick with the separated results and discussion format, to be able to discuss all potential drivers in a step-wise manner. We carefully edited the manuscript to minimize the redundancies between results and discussion.**

Specific comments:

Figure 1: please, add the catchment areas on the Map by indicating the Site code.

**We added the site code in Figure 1 in the revised version of the manuscript.**

Lines 216-221: Please, explain better how you have split your dataset. It is not fully clear to me.

**We updated the explanation of splitting the dataset accordingly.**

Lines 320-321: Have you considered a delayed input or a direct input (von Freyberg et al. 2018) for estimating new water in snow-dominated catchments? I think that should be discussed more about the role of snowpack on the estimation of the new water and what are the consequences of choosing a direct or delayed input on new water fraction (similarly to what von Freyberg et al. 2018 did for young water fraction).

**Thank you for this remark. In the revised version of the manuscript, we updated the discussion on the impact of delayed snowmelt on young and new waters.**

Lines 505-508: Are there previous studies that support this result? If not, please provide an explanation to this counterintuitive result.

**We now explain the potential reasoning (i.e., cross-correlation with elevation and steeper slopes) for this in more detail in the revised version of the manuscript.**

Table 3: Add the fields "min elevation" and "max elevation"

**We added "min" elevation (the location of the gauge) in Table 3, "max. elevation" can be inferred from the already available column "elevation difference".**

Figure 5b: Please add the site code in panel b. If possible, think about a possible alternative representation of Figure 5 since it is really intricate.

**Thank you, we understand that the representation is not ideal, however upon discussion we did not find a more suitable way to present this. However, we believe that the important overall**

**message of the figure is that only in roughly half of the catchments F$_{new}$ substantially increases with increasing precipitation, thus we moved panel (b) of the Figure to the supplementary material (now in the Supplementary Material in Figure S2).**

Figure 7 & Figure 8: I would suggest to represent rainfall-dominated, hybrid and snow-dominated catchments by using three different colors. This could lead to additional insights to improve the discussion about these results.

**Thank you for this suggestion, we updated the Figures accordingly and now show the different precipitation regimes in different colors.**

---

## Author Response (AR2)

**We thank the reviewer for the interesting comments and suggestions. Below we provide a detailed point-by-point response (in bold) to the reviewers' comments (*in italic*).**
* * *
*The authors have responded to most my concerns. I understand that substantial or anticipated conclusions were always restricted by limited observations of isotopes in ALPs. However, I partly don't agree that the explanations below about why high fractions of new water (Fnew) were more likely in small catchments.*

**Thank you for acknowledging the implemented changes. We additionally address the final issues raised in the revised version of the manuscript.**

*First, high-elevation catchments have greater subsurface storage leading to longer transit times. However, it is widely reported that rainfall can directly drain to drainage networks as new fractions in high altitude areas in alpine basins. While, in the few existing studies mainly located in lower relief basins as shown in the red circled part of FIGURE 10, regolith seems to be thick to store more water as rain falls (Grant et al., 2017; McCormick et al., 2021).*

**We now expanded the discussion and added a sentence on the importance of regolith storage: "Previous studies also reported that weathered bedrock or regolith is important in storing water and delaying runoff response (Grant and Dietrich, 2017; McCormick et al., 2021) by enhancing storage capacity especially in steeper catchments where precipitation would otherwise drain to river networks directly."**

*Second, I agree with Hrachowitz et al. (2021) findings. Many classical studies also verified that new water tends to percolate deeply and quickly to mix with old matrix soil water in preferential-pathway-developed forest catchments (see in Weiler et al., 2005), which may lead to more old water in stormflow. Hence, I argue that the authors should provide more convincing discussions.*

**Thank you. We now updated the discussion on the effects of forest in the according section of the revised manuscript. "The formation of preferential pathways can be argued to lead to either more or less recent precipitation ending up in streamflow. Preferential pathways may transport precipitation directly to streams, thus**

increasing Fnew and Fyw (Brantley et al., 2017; von Freyberg et al., 2018). However, previous studies also argued that preferential pathways tend to increase deep percolation and mixing in deeper storages, thus decreasing Fnew and Fyw (Hrachowitz et al. 2021; Weiler et al., 2006). It remains to be tested whether the correlations between Fnew and the fraction of catchment area covered by forests might also be an artefact of cross-correlations with other variables."

References:

Grant, G. E., and W. E. Dietrich (2017), The frontier beneath our feet, Water Resour. Res., 53, 2605–2609, doi:10.1002/2017WR020835.

Hrachowitz, M., Stockinger, M., Coenders-Gerrits, M., van der Ent, R., Bogena, H., Lücke, A., and Stumpp, C.: Reduction of vegetation-accessible water storage capacity after deforestation affects catchment travel time distributions and increases young water fractions in a headwater catchment, Hydrol. Earth Syst. Sci., 25, 4887–4915, https://doi.org/10.5194/hess-25 4887-2021, 2021.

McCormick, E. L., Dralle, D. N. & Hahm, W. J., et al. (2021). Widespread woody plant use of water stored in bedrock. Nature, 597(7875), 225-229.

Weiler, M., McDonnell, J.J., Tromp-van Meerveld, I., & Uchida, T.(2005). Subsurface stormflow. In M. G. Anderson & J. J. McDon-nell (Eds.),Encyclopedia of hydrological sciences(S. hsa119). JohnWiley & Sons, Ltd.. https://doi.org/10.1002/0470848944. hsa119.